# Adaptive Test-Time Personalization for Federated Learning

**Wenxuan Bao**[1*], **Tianxin Wei**[1*], **Haohan Wang**[1], **Jingrui He**[1]
[1]University of Illinois Urbana-Champaign
{wbao4,twei10,haohanw,jingrui}@illinois.edu

## Abstract

Personalized federated learning algorithms have shown promising results in adapting models to various distribution shifts. However, most of these methods require labeled data on testing clients for personalization, which is usually unavailable in real-world scenarios. In this paper, we introduce a novel setting called test-time personalized federated learning (TTPFL), where clients locally adapt a global model in an unsupervised way without relying on any labeled data during test-time. While traditional test-time adaptation (TTA) can be used in this scenario, most of them inherently assume training data come from a single domain, while they come from multiple clients (source domains) with different distributions. Overlooking these domain interrelationships can result in suboptimal generalization. Moreover, most TTA algorithms are designed for a specific kind of distribution shift and lack the flexibility to handle multiple kinds of distribution shifts in FL. In this paper, we find that this lack of flexibility partially results from their pre-defining which modules to adapt in the model. To tackle this challenge, we propose a novel algorithm called `ATP` to adaptively learns the adaptation rates for each module in the model from distribution shifts among source domains. Theoretical analysis proves the strong generalization of `ATP`. Extensive experiments demonstrate its superiority in handling various distribution shifts including label shift, image corruptions, and domain shift, outperforming existing TTA methods across multiple datasets and model architectures. Our code is available at `https://github.com/baowenxuan/ATP`.

## 1  Introduction

Federated learning (FL) is a distributed learning system where multiple clients collaborate to train a machine learning model under the orchestration of the central server, while keeping their data decentralized [31, 18]. However, clients in FL typically exhibit distinct data distributions. For example, in the context of animal image classification, users tend to capture pictures of various animals prevalent in their respective regions, introducing label shift [51] to the local image dataset. Meanwhile, even when capturing images of the same species, the visual appearance can be influenced by the environment and camera settings, introducing feature shift [34]. It is crucial that each client can adapt the model to align with its unique data distribution [44]. Previous personalized federated learning (PFL) works have mainly focused on improving the performance on clients participating in training [41, 37, 25, 4] or generalization to new clients [8, 7, 6], assuming the availability of labeled data. However, in many real-world scenarios, clients do not have labeled data for personalization, which limits the application of PFL algorithms. For example, when employing an animal image classifier to mobile phones, their users may capture images of various animals, but without any accompanying labels indicating the species of the animal.

---

*Equal contribution.

37th Conference on Neural Information Processing Systems (NeurIPS 2023).

In this paper, we introduce a novel setting named test-time personalized federated learning (TTPFL). During the training phase, a global model is trained using source clients. During the testing phase, each target client downloads the global model and locally personalizes the model with its unlabeled data during test-time. This setting is particularly well-suited for cross-device FL, especially when generalizing to a large number of target clients that have not participated in the training phase and lack labeled data for supervised personalization. Compared to global FL, which trains a shared global model for all clients, TTPFL enables model adaptation to individual target clients facing complex distribution shifts. Compared to standard PFL, TTPFL does not necessitate additional labeled data from target clients for adaptation.

Test-time adaptation (TTA), which adapts a pretrained model from the source domain to an unlabeled target domain, could be a solution for TTPFL. However, applying current TTA methods to FL poses two challenges. First, most TTA methods assume training data are sampled from a single domain [16, 52]. In FL, where source data are distributed across multiple clients, this simplification neglects interrelationships among source domains, impacting generalization. Furthermore, the current TTA methods are usually customized for specific distribution shifts and lack the flexibility to address diverse types of distribution shifts in FL. The inflexibility of existing TTA algorithms largely results from their predefined selection of modules to adapt (e.g., feature extractor [28, 43], final linear layer [16, 36], batch normalization layers [38, 45]). However, different modules encode varying semantic information levels, and adapting specific modules may be effective for certain shifts but not others [20]. Meanwhile, although the distribution shifts among source and target clients cannot be directly inspected, the same type of distribution shifts is likely to exist among source clients. We argue that

> *Which modules to adapt should depend on the type of distribution shifts among clients, which can be inferred from source clients.*

Motivated by this, we propose a new Adaptive Test-time Personalization algorithm called `ATP` to learn the adaptation rates from distribution shifts among source clients. During training, each source client simulates unsupervised adaptation and refine the adaptation rates of each module to maximize the effect of unsupervised adaptation. The server aggregates local adaptation rates periodically to improve generalization. During testing, each target client leverages learned adaptation rates to locally adapt the global model, and cumulatively averages adapted models from previous batches to enhance the performance for online TTA. Theoretical analysis confirms `ATP`'s robust generalization due to its utilization of multiple sources and low-dimensional adaptation rates. Extensive experiments demonstrate its superiority in addressing various distribution shifts scenarios, including label shift, image corruptions, and domain shift, consistently outperforming existing TTA methods across multiple datasets and model architectures. *We summarize our contributions as follows.*

- We consider TTPFL, a new learning setting in FL, addressing the challenge of generalizing to new unlabeled clients under complex distribution shifts. (Section 3)
- We introduce `ATP`, which adaptively learns the adaptation rate for each module, enabling it to handle different types of distribution shifts. (Section 4)
- We provide theoretical analysis confirming `ATP`'s robust generalization. (Section 5)
- We empirically evaluate `ATP` over various distribution shifts scenarios, using a wide range of datasets and models. (Section 6)

## 2   Related works

**Federated learning** (FL) is a distributed learning system where multiple clients collaborate to train a machine learning model under a central server's orchestration while keeping data decentralized [18].

**Personalized federated learning** (PFL) extends this framework by allowing each client to personalize the model to its own local data. The most straightforward PFL method is fine-tuning the global model with a few steps of gradient descent [48, 8, 7]. Similarly, another line of works use the global model as a regularizer [22] during local training. FedTHE [17] focuses on evolving local testing set, and proposes a test-time adaptation algorithm for FL that adaptively combines global and personalized models. However, all these methods require labeled data to construct personalized models. Fed-RoD [6] uses hypernetworks to generate personalized model, relaxing the requirements for labeled data. But it still requires the label distribution of the client. FedUL [30] trains a global model with only

unlabeled clients. However, it is limited to label shift where each client shares the label-conditional feature distribution $p(\boldsymbol{x}|\boldsymbol{y})$. Our setting is mostly similar to OD-PFL [2], which also focuses on generalization to new unlabeled client. It uses an unsupervised client encoder and a hypernetwork [39] to generate personalized model. However, OD-PFL requires re-training a large hypernetwork, while our TTPFL setting focuses on adapting an existing global model.

**Test-time adaptation** (TTA) aims to adapt a machine learning model to a testing set with dataset shift during test-time without re-accessing training data. Most of the TTA methods focus on either feature shift or label shift. For feature shift (same $p(\boldsymbol{y}|\boldsymbol{x})$, different $p(\boldsymbol{x})$), entropy minimization is frequently used to adapt the model in the unsupervised fashion. Tent [45] minimizes the average prediction entropy by adapting the batch normalization layers [15]. MEMO [52] minimizes the marginal entropy over different augmentations of the sample input image by adjusting all model parameters. SHOT [28] exploits information maximization and pseudo-labeling to achieve target-specific feature extraction. Differently, T3A [16] adjusts the final classification layer, but it is also shown to implicitly reduce the entropy. It is important to notice that all these methods pre-define which modules to be adapted in the network. For label shift (same $p(\boldsymbol{x}|\boldsymbol{y})$, different $p(\boldsymbol{y})$), most of the previous works focus on estimating the shifted label distribution. EM [36, 1] iteratively uses model predictions to estimate the label prior distribution and uses label prior distribution to adjust model predictions. BBSE [29, 3] constructs a confusion matrix on the validation dataset, and uses the prediction distribution to estimate the ground-truth label distribution. The estimated label distribution is used for re-training a model with importance sampling. [49] generalizes these methods to the online dataset shift setting where the label distribution for testing data is evolving over time. However, all these methods heavily rely on the assumption of the same $p(\boldsymbol{x}|\boldsymbol{y})$, which can be violated in real applications.

**Comparison with FedTHE [17]**  Recently, FedTHE also explored TTA in FL. However, FedTHE focus on the test-time distribution shift for clients that participate in FL training, while we focus on improving the performance on novel clients. Moreover, FedTHE fuses global head and personalized head to get robust prediction. It cannot be easily generalized to target clients which does not have labeled data to train the personalized head.

Our paper is also related to partial fine-tuning and hyperparameter optimization. We discuss these works in Appendix A.1 in detail.

## 3  Motivation

In this section, we first introduce the setting of test-time personalized federated learning, and then show that current TTA methods lack the flexibility to various types of distribution shifts in TTPFL.

### 3.1  Test-time personalized federated learning

**Preliminary**  We consider a standard setting for cross-device FL [46] and domain generalization [47]. Considering an FL system with $N$ source clients $\{\mathcal{S}_i\}_{i=1}^N$ and $M$ target clients $\{\mathcal{T}_j\}_{j=1}^M$. Each source client $\mathcal{S}_i$ has its own *labeled* dataset $\mathbb{D}^{\mathcal{S}_i}$ with $n_i$ samples $\{(\boldsymbol{x}_1^{\mathcal{S}_i}, \boldsymbol{y}_1^{\mathcal{S}_i}), \cdots, (\boldsymbol{x}_{n_i}^{\mathcal{S}_i}, \boldsymbol{y}_{n_i}^{\mathcal{S}_i})\}$ i.i.d. drawn from its distribution $P^{\mathcal{S}_i}(\boldsymbol{x}, \boldsymbol{y})$, where $\boldsymbol{x}$ is the input and $\boldsymbol{y}$ is its corresponding label. Each target client $\mathcal{T}_j$ has its own *unlabeled* dataset $\mathbb{X}^{\mathcal{T}_j} = \{\boldsymbol{x}_1^{\mathcal{T}_j}, \cdots, \boldsymbol{x}_{m_j}^{\mathcal{T}_j}\}$ i.i.d. drawn from its distribution $P^{\mathcal{T}_j}(\boldsymbol{x}, \boldsymbol{y})$, while the corresponding labels $\{\boldsymbol{y}_1^{\mathcal{T}_j}, \cdots, \boldsymbol{y}_{m_j}^{\mathcal{T}_j}\}$ cannot be accessed. The distributions for different source/target clients are different, sampled from a meta-distribution $\mathcal{Q}$, i.e., distribution of distributions. *Global federated learning* (GFL) aims to find a single global model minimizing the expected loss over client population [46]:

$$\mathcal{L}(\boldsymbol{w}_G) = \mathbb{E}_{P \sim \mathcal{Q}} \mathcal{L}_P(\boldsymbol{w}_G), \text{ where } \mathcal{L}_P(\boldsymbol{w}_G) = \mathbb{E}_{(\boldsymbol{x}, \boldsymbol{y}) \in P} \ell(f(\boldsymbol{x}; \boldsymbol{w}_G); \boldsymbol{y}) \tag{1}$$

where $\ell$ represents the loss function and $f$ represents model. GFL enforces that each client uses the same global model for prediction, which does not allow for adaptation to each client's unique data distribution. In contrast, *personalized federated learning* (PFL) personalizes the global model $\boldsymbol{w}_G$ using its labeled data, and uses the personalized model for prediction, replacing the $\boldsymbol{w}_G$ in Eq. (1). However, most of the PFL algorithms [8, 7, 22] require the assumption that the target client also possesses additional labeled data, which is a stronger assumption compared to GFL.

**Test-time personalized federated learning**  In this paper, we introduce a novel setting named *test-time personalized federated learning* (TTPFL), and compare it with the standard GFL and PFL

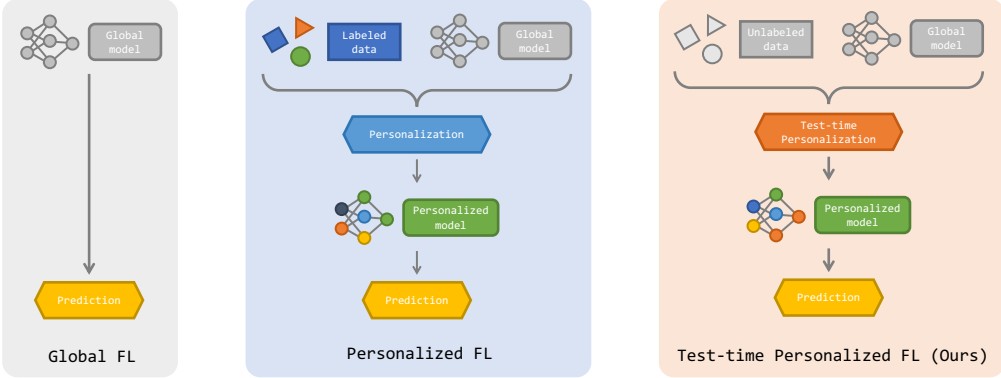

Figure 1: Comparison between the testing phase of GFL, PFL, and TTPFL. TTPFL enables model personalization without requiring labeled data.

in Figure 1. TTPFL focuses on how to adapt a trained global model to each target client's data distributions during *test-time*, with an adaptation rule $\mathcal{A}$ only using unlabeled data. The objective function can be formulated as

$$\mathcal{L}(\boldsymbol{w}_G, \mathcal{A}) = \mathbb{E}_{P \sim \mathcal{Q}} \mathcal{L}_P(\boldsymbol{w}_G, \mathcal{A}), \text{ where } \mathcal{L}_P(\boldsymbol{w}_G, \mathcal{A}) = \mathbb{E}_{(\boldsymbol{x}, \boldsymbol{y}) \in P} \ell(f(\boldsymbol{x}; \mathcal{A}(\boldsymbol{w}_G, \boldsymbol{X})); \boldsymbol{y}) \quad (2)$$

which can be unbiasedly estimated by the average loss over $M$ target clients unseen during training

$$\hat{\mathcal{L}}(\boldsymbol{w}_G, \mathcal{A}) = \frac{1}{M} \sum_{j=1}^{M} \hat{\mathcal{L}}_{P^{\mathcal{T}_j}}(\boldsymbol{w}_G, \mathcal{A}), \text{ where } \hat{\mathcal{L}}_{P^{\mathcal{T}_j}}(\boldsymbol{w}_G, \mathcal{A}) = \frac{1}{m_j} \sum_{r=1}^{m_j} \ell(f(\boldsymbol{x}_r^{\mathcal{T}_j}; \mathcal{A}(\boldsymbol{w}_G, \boldsymbol{X}_r^{\mathcal{T}_j})); \boldsymbol{y}_r^{\mathcal{T}_j}) \quad (3)$$

The adaptation rule $\mathcal{A}$ adapts a the global model with unlabeled samples $\boldsymbol{X}_r^{\mathcal{T}_j}$. We consider two standard settings: *test-time batch adaptation* (TTBA) and *online test-time adaptation* (OTTA) [27]. TTBA individually adapts the global model to each batch of unlabeled samples, where $\boldsymbol{X}_r^{\mathcal{T}_j}$ is the data batch that $\boldsymbol{x}_r^{\mathcal{T}_j}$ belongs. OTTA adapts the global model in an online manner, where $\boldsymbol{X}_r^{\mathcal{T}_j}$ contains all the data batches arriving before or together with $\boldsymbol{x}_r^{\mathcal{T}_j}$.

## 3.2 Limitation of test-time adaptation

As the precursor to TTPFL, TTA [45, 52, 29] studies how to adapt a trained model to target dataset under certain types of dataset shifts. Since TTA methods only require unlabeled target data for adaptation, they can be applied in TTPFL. We test state-of-the-art TTA methods with ResNet-18 on CIFAR-10 under two types of distribution shifts: label shift and feature shift, with results presented in Figure 2. As expected, each algorithm can boost the model's accuracy under the distribution shift it is designed for. However, most algorithms improve their performance in one scenario while simultaneously impairing it in another scenario, demonstrating a trade-off in their performance on feature shift and label shift. Moreover, when facing a more complex hybrid of distribution shifts, most TTA methods fail to introduce satisfactory performance gain (Table 1). Therefore, TTA methods are not suitable for TTPFL given the variety of distribution shifts in FL client.

The inflexibility of TTA algorithms largely results from their predefined selection of modules to adapt, e.g., batch normalization (BN) layers [38, 45], the feature extractor [28, 43], or the last linear layer [16, 36]. However, which modules to adapt is closely related to the type of distribution shift. For example, adapting the last linear layer can encode the label shift (Proposition 3.1), while it may fail when the extracted features are already corrupted due to feature shift. Similarly, adapting the BN layers can improve the performance under feature shift by distribution alignment (Proposition 3.2), while distribution alignment can harm the performance under label shift [53].

**Proposition 3.1** (Adapting the last layer to handle label shift)**.** *Consider two distribution $p, q$ with $p(\boldsymbol{x}|\boldsymbol{y}) = q(\boldsymbol{x}|\boldsymbol{y})$ and $p(\boldsymbol{y}) \neq q(\boldsymbol{y})$. When a neural network is calibrated on $p$, i.e., $f(\boldsymbol{x}; \boldsymbol{w}) = p(\cdot|\boldsymbol{x})$, it is calibrated on $q$ after adding $\log \frac{q(\boldsymbol{y})}{p(\boldsymbol{y})}$ to the bias term of the final last layer.*

**Proposition 3.2** (Adapting the BN layer to handle feature shift [38])**.** *When the feature shift only causes differences in the first and second order moments of the feature activations $\boldsymbol{z} = g(\boldsymbol{x})$ where $g$*

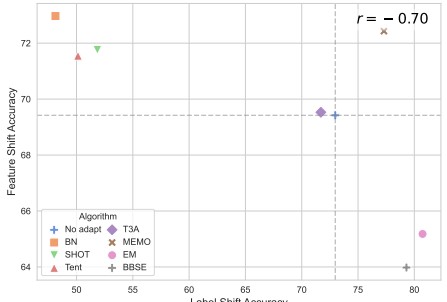
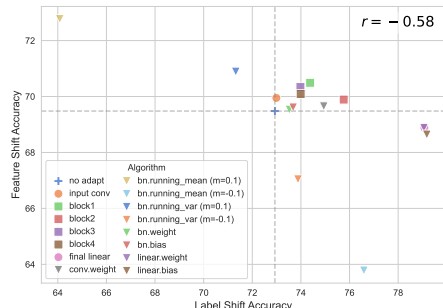

Figure 2: Performance trade-off of existing TTA methods under two distribution shifts.

Figure 3: Performance trade-off of entropy minimization when adapting different modules.

*is the combination of layers before the BN layer, the feature shift can be removed by adapting running mean and variance of the BN layer.*

To verify the connection between distribution shift and the selection of modules for adaptation, we experiment with adapting different subsets of modules within the network to minimize the entropy loss [45]. In Figure 3, we observe a similar performance trade-off between feature shift and label shift: while adapting certain modules can boost the accuracy under one distribution shift, it is less likely to succeed under the other shift. To break the performance trade-off, it is essential to adaptively choose which modules to adapt according to the present type of distribution shift. Moreover, while [20] suggests adapting different blocks in the network, we find it more important to decide (1) which module type to adapt and (2) what is the adaptation rate (i.e., learning rate for adaptation). For example, adapting all BN running means significantly outperforming adapting any one block under feature shift. Meanwhile, employing positive or negative adaptation rates for running means yields contrasting outcomes, favoring adaptation in the presence of label shift or feature shift while impairing the other. These observations motivate us to choose which module to adapt (instead of blocks) while optimizing the adaptation rates for each module.

## 4 `ATP`: adaptive test-time personalization

In this section, we propose `ATP` that automatically learns the adaptation rates for each module. We introduce the training and testing phase of `ATP` in subsection 4.1 and 4.2, respectively.

### 4.1 Training phase: learn to adapt with source clients

In this part, we introduce how `ATP` learns adaptation rates from source clients without sharing local data. `ATP` uses the communication protocol of FedAvg [31] to optimize adaptation rates. In each communication round, each source client first simulates unsupervised adaptation with the current adaptation rates, and then refines the adaptation rates to maximize the effect of adaptation. After local computation, the local adaptation rates are then aggregated on the server to ensure better generalization to target clients. Algorithm 1 gives the overview of the training phase of `ATP`. We then explain each step in detail.

**Unsupervised adaptation** We consider a neural network model $f(\cdot; \boldsymbol{w}_G)$ with global model parameter $\boldsymbol{w}_G \in \mathbb{R}^D$. Similar to previous works [45, 38], we consider the model processes a data batch $\boldsymbol{X}_k^{\mathcal{S}_i} = \{\boldsymbol{x}_{k,b}^{\mathcal{S}_i}\}_{b=1}^B$ at a time where $B$ is the batch size, $i$ is the client index and $k$ is the batch index. In the following, we omit the superscript $\mathcal{S}_i$ for clarity, e.g. $\boldsymbol{X}_k^{\mathcal{S}_i} \to \boldsymbol{X}_k$, as unsupervised adaptation and supervised refinement operate identically across all source clients. The network has $d$ modules, with corresponding parameters $\boldsymbol{w}^{[1]}, \cdots, \boldsymbol{w}^{[d]}$. Typically we have $d \ll D$. During unsupervised adaptation, we allow each module $\boldsymbol{w}^{[l]}$ to have a different adaptation rate $\alpha^{[l]}$. `ATP` learns to adapt both trainable parameters and running statistics for batch normalization (BN) [15] layers. To achieve more precise control of adaptation, the 'module' in `ATP` is slightly more fine-grained than the 'layer'. For example, each BN layer has four modules: running mean, running variance, weight, and bias.

*Update trainable parameters*  A common strategy for updating trainable parameters is performing one step of gradient descent to minimize the cross-entropy loss. Since label information are unavailable for computing cross-entropy, we instead minimize the entropy loss $\ell_H(\hat{\boldsymbol{Y}}) = \frac{1}{B}\sum_{b=1}^{B}(-\sum_c \hat{y}_{b,c}\log\hat{y}_{b,c})$, where $\hat{\boldsymbol{Y}}$ is the prediction probabilities over the label space of a data batch. Entropy quantifies the uncertainty of the model prediction, and is frequently used in previous TTA algorithms [45, 52, 28]. For each trainable parameter module $\boldsymbol{w}^{[l]}$, the corresponding unsupervised update direction for each client is the negative gradient direction, i.e.,

$$\boldsymbol{h}_k^{[l]} = -\nabla_{\boldsymbol{w}^{[l]}}\ell_H(f(\boldsymbol{X}_k;\boldsymbol{w}_G)) \tag{4}$$

*Update running statistics*  The running statistics (mean/variance) in BN layers are not updated by gradient descent. Instead, they are updated by running average.

$$\boldsymbol{w}_k^{[l]} \leftarrow (1-m)\boldsymbol{w}_G^{[l]} + m\hat{\boldsymbol{w}}_k^{[l]} = \boldsymbol{w}_G^{[l]} + m(\hat{\boldsymbol{w}}_k^{[l]} - \boldsymbol{w}_G^{[l]})$$

where $\boldsymbol{w}_G^{[l]}$ is the running statistics and $\hat{\boldsymbol{w}}_k^{[l]}$ is the statistic for the current batch of inputs. In previous works, the momentum[1] $m$ is usually a fixed hyperparameter in $[0,1]$. In ATP, we consider the momentum for each module as an adaptation rate ($\alpha^{[l]} \in \mathbb{R}$) to be learned. We define the corresponding update direction as

$$\boldsymbol{h}_k^{[l]} = \hat{\boldsymbol{w}}_k^{[l]} - \boldsymbol{w}_G^{[l]} \tag{5}$$

After computing the update direction, each module will be updated along the update direction with its corresponding adaptation rate, i.e., $\boldsymbol{w}_k^{[l]} \leftarrow \boldsymbol{w}_G^{[l]} + \alpha^{[l]}\boldsymbol{h}_k^{[l]}$. Expressed in a compact form,

$$\boldsymbol{w}_k \leftarrow \boldsymbol{w}_G + (\boldsymbol{A}\boldsymbol{\alpha}) \odot \boldsymbol{h}_k \tag{6}$$

where $\odot$ is the element-wise product, $\boldsymbol{h}_k \in \mathbb{R}^D$ is the concatenation of $\{\boldsymbol{h}_k^{[l]}\}_{l=1}^d$, $\boldsymbol{\alpha} = [\alpha^{[1]}, \cdots, \alpha^{[d]}]^\top$ and $\boldsymbol{A} \in \mathbb{R}^{D\times d}$ is a 0-1 assignment matrix that maps each adaptation rate $\alpha^{[l]}$ to the indices of $l$-th module's parameters in $\boldsymbol{w}_G$.

**Supervised refinement**  After unsupervised adaptation, we refine the adaptation rates on each source client with label information to minimize $\ell_{CE}(f(\boldsymbol{X}_k, \boldsymbol{w}_k), \boldsymbol{Y}_k)$, where $\ell_{CE}$ is the cross-entropy loss. We use gradient descent to optimize $\boldsymbol{\alpha}$, i.e.,

$$\boldsymbol{\alpha} \leftarrow \boldsymbol{\alpha} - \eta\nabla_{\boldsymbol{\alpha}}\ell_{CE}(f(\boldsymbol{X}_k;\boldsymbol{w}_k), \boldsymbol{Y}_k) \tag{7}$$

where $\eta$ is the learning rate of adaptation rates. Notice that the gradient of $\boldsymbol{\alpha}$ can be computed as

$$\nabla_{\boldsymbol{\alpha}}\ell_{CE}(f(\boldsymbol{X}_k;\boldsymbol{w}_k), \boldsymbol{Y}_k) = \frac{\partial\ell_{CE}(f(\boldsymbol{X}_k;\boldsymbol{w}_k), \boldsymbol{Y}_k)}{\partial\boldsymbol{w}_k}\frac{\partial\boldsymbol{w}_k}{\partial\boldsymbol{\alpha}} = \boldsymbol{A}^\top(\boldsymbol{h}_k \odot \nabla_{\boldsymbol{w}_k}\ell_{CE}(f(\boldsymbol{X}_k;\boldsymbol{w}_k), \boldsymbol{Y}_k))$$

To estimate the gradient of $\boldsymbol{\alpha}$, each training client only needs to adjacently compute the unsupervised and supervised gradient, and compute their module-wise inner products. Different from many meta-learning algorithms [9, 26], ATP is computationally very efficient since it requires no second-order derivatives. In the practical implementation, since each module in the model has significantly different number of parameters, the raw gradient for each $\alpha^{[l]}$ usually has different scales. Therefore we normalize the gradient with the square root of the number of parameters in the corresponding module.

**Server aggregation**  To incorporate adaptation knowledge from multiple source clients and enhance generalization to the clients' population, ATP use standard federated aggregation [31] to periodically aggregates the local adaptation rates. In each communication rounds, after each client locally update $\boldsymbol{\alpha}$ for a few iterations, the local adaptation rates are uploaded to the server for averaging (as shown in line 6 of Algorithm 1), and then sent to source clients for the next round of training. With server aggregation, ATP learn the adaptation rates that enables successful adaptation to all source clients in average.

*Communication cost*  Notice that ATP only optimizes the adaptation rates $\boldsymbol{\alpha}$ without changing the global model $\boldsymbol{w}_G$. Therefore, only the adaptation rates are kept transmitted between the server and each client, while the global model parameter is only broadcasted once at the start of the ATP training. Such design significantly reduces the communication cost from $2TD$ (for standard FedAvg) to $D + 2Td$.

---

[1]Some literatures consider $(1-m)$ as the momentum. Here we follow the definition in PyTorch.

**Algorithm 1** ATP Training

```
ServerTrain(w_G, α_G^0 = 0)
```
1: Broadcast $\boldsymbol{w}_G$ to all source clients
2: **for** communication round $t = 1$ to $T$ **do**
3:    $\mathbb{S}^t \leftarrow$ (random set of $C$ source clients)
4:    **for** source client $\mathcal{S}_i \in \mathbb{S}^t$ **in parallel do**
5:       $\boldsymbol{\alpha}_i^t \leftarrow$ ClientTrain($\mathcal{S}_i, \boldsymbol{\alpha}_G^{t-1}$)
6:       $\boldsymbol{\alpha}_G^t = \frac{1}{C} \sum_{\mathcal{S}_i \in \mathbb{S}^t} \boldsymbol{\alpha}_i^t$
7: **return** $\boldsymbol{\alpha}_G^T$

```
ClientTrain(S_i, α)        # Run on source client S_i
```
8: **for** local epoch $e = 1$ to $E$ **do**
9:    $\mathbb{B}^{\mathcal{S}_i} \leftarrow$ (split $\mathbb{D}^{\mathcal{S}_i}$ into $K^{\mathcal{S}_i}$ batches of size $B$)
10:    **for** batch $k = 1$ to $K^{\mathcal{S}_i}$ **do**
11:       $(\boldsymbol{X}_k^{\mathcal{S}_i}, \boldsymbol{Y}_k^{\mathcal{S}_i}) \leftarrow$ ($k$-th labeled batch in $\mathbb{B}^{\mathcal{S}_i}$)
12:       Estimate update direction $\boldsymbol{h}_k^{\mathcal{S}_i}$ with *unlabeled* $\boldsymbol{X}_k^{\mathcal{S}_i}$ according to Eq. (4) and (5)
13:       $\boldsymbol{w}_k^{\mathcal{S}_i} \leftarrow \boldsymbol{w}_G + (\boldsymbol{A}\boldsymbol{\alpha}) \odot \boldsymbol{h}_k^{\mathcal{S}_i}$
14:       $\boldsymbol{\alpha} \leftarrow \boldsymbol{\alpha} - \eta \nabla_{\boldsymbol{\alpha}} \ell_{CE}(f(\boldsymbol{X}_j^{\mathcal{S}_i}; \boldsymbol{w}_k^{\mathcal{S}_i}), \boldsymbol{Y}_k^{\mathcal{S}_i})$
15: **return** $\boldsymbol{\alpha}$

**Algorithm 2** ATP Testing

```
ClientTest(T_j, w_G, α)   # Run on target client T_j
```
1: $\mathbb{B}^{\mathcal{T}_j} \leftarrow$ (split $\mathbb{X}^{\mathcal{T}_j}$ into $K^{\mathcal{T}_j}$ batches of size $B$)
2: $\boldsymbol{h}_{\text{history}} \leftarrow \boldsymbol{0}$       *# Cumulative moving average*
3: **for** batch $k = 1$ to $K^{\mathcal{T}_j}$ **do**
4:    Estimate update direction $\boldsymbol{h}_k^{\mathcal{T}_j}$ with *unlabeled* $\boldsymbol{X}_k^{\mathcal{T}_j}$ according to Eq. (4) and (5)
5:    **if** *TTBA* **then**
6:       $\boldsymbol{w}_k^{\mathcal{T}_j} \leftarrow \boldsymbol{w}_G + (\boldsymbol{A}\boldsymbol{\alpha}) \odot \boldsymbol{h}_k^{\mathcal{T}_j}$
7:    **else if** *OTTA* **then**
8:       $\boldsymbol{h}_{\text{history}} \leftarrow \frac{k-1}{k} \boldsymbol{h}_{\text{history}} + \frac{1}{k} \boldsymbol{h}_k^{\mathcal{T}_j}$
9:       $\boldsymbol{w}_k^{\mathcal{T}_j} \leftarrow \boldsymbol{w}_G + (\boldsymbol{A}\boldsymbol{\alpha}) \odot \boldsymbol{h}_{\text{history}}$
10:    Make prediction: $\hat{\boldsymbol{Y}}_k^{\mathcal{T}_j} = f(\boldsymbol{X}_k^{\mathcal{T}_j}; \boldsymbol{w}_k^{\mathcal{T}_j})$

## 4.2 Testing phase: exploit adaptation rates on target clients

During testing, each target client downloads both the global model and the adaptation rates. We propose two versions of ATP: ATP-batch for test-time batch adaptation (TTBA) and ATP-online for online test-time adaptation (OTTA). We summarize the testing phase in Algorithm 2.

ATP-**batch** For TTBA, each target client makes independent predictions on each batch. For each batch of target data, ATP-batch first conducts the unsupervised adaptation identical to source clients, and then makes prediction.

ATP-**online** For OTTA, data comes in a stream of batches $[\boldsymbol{X}_1^{\mathcal{T}_j}, \boldsymbol{X}_2^{\mathcal{T}_j}, \cdots]$. Previous works [43, 45] usually keep updating the model batch after batch. However, such accumulative adaptation can introduce severe batch dependency problem, i.e., each batch is evaluated when the model takes different number of update steps [54]. For the first few batches, the model has not adapted to the local distribution well; while for the last few batches, the model may over-minimize the entropy but increase the cross-entropy loss. To avoid batch dependency, we propose an *averaged* adaptation mechanism for online adaptation, whose scale of adaptation is stable during online adaptation.

For each batch $\boldsymbol{X}_k^{\mathcal{T}_j}$ in the data stream, we always compute the update direction $\boldsymbol{h}_k^{\mathcal{T}_j}$ starting with the fixed global model $\boldsymbol{w}_G$ according to Eq. (4) and (5). Subsequently, instead of using only the current update direction to adapt the model, we *average* all the stored update direction to update the model, i.e.,

$$\boldsymbol{w}_k^{\mathcal{T}_j} \leftarrow \boldsymbol{w}_G + (\boldsymbol{A}\boldsymbol{\alpha}) \odot \left( \frac{1}{k} \sum_{s=1}^k \boldsymbol{h}_s^{\mathcal{T}_j} \right) \tag{8}$$

By using the average of previous updates, we simulate updating with larger batch size to utilize historical data, while controlling the number of update steps to be one. In the practical implementation, we use cumulative moving average (as shown in line 8 of Algorithm 2), whose space complexity does not increase with the increment of step $k$.

## 5 Theoretical analysis

In this section, we show that ATP enjoys good generalization guarantees because of the low dimensionality of adaptation rates. Formal definitions, assumptions and full proofs are provided in Appendix B.3. We also show in Appendix B.2 that ATP has convergence guarantee similar to FedAvg [31, 46].

**Theorem 5.1** (Generalization)**.** *Let* $\mathcal{H} = \{\boldsymbol{\alpha} : \|\boldsymbol{\alpha}\|_2 \leq R\}$ *be the hypothesis space (space of adaptation rates),* $N$ *be the number of source clients, and* $K$ *be the number of data batches on each*

*source client. Assuming (1) L-Lipschitz model, and (2) H-upper-bounded 2-norms for each module's update. For any fixed global model $\boldsymbol{w}_G$ and any $\epsilon > 0$, we have*

$$\Pr(\sup_{\boldsymbol{\alpha} \in \mathcal{H}} |\varepsilon(\boldsymbol{\alpha}) - \hat{\varepsilon}(\boldsymbol{\alpha})| \geq \epsilon) \leq \left( \frac{12LHR}{\epsilon} \right)^d \cdot 4 \exp \left( -\frac{NK\epsilon^2}{2(\sqrt{K}+1)^2} \right) \tag{9}$$

*where $\hat{\varepsilon}(\boldsymbol{\alpha})$ is the average **post-adaptation** error rate on source clients, and $\varepsilon(\boldsymbol{\alpha})$ is the expected **post-adaptation** error rate on clients' population.*

Theorem 5.1 shows that, although `ATP` improves the model expressiveness by adapting the model to each client's distribution, `ATP` can still provably generalize well to the clients' population. Especially, this generalization benefit from low dimensionality of adaptation rates, since the bound get looser when $d$ increases. Moreover, this bound shows the importance of learning adaptation rates from multiple source clients: if we merge all $N$ source domains with $K$ batches into one domain with $NK$ batches, then the bound will be much looser.

## 6 Experiments

In this section, we design experiments to answer the following research questions:

- **RQ1**: Can `ATP` handle different distribution shift and outperform prior TTA methods?
- **RQ2**: Does `ATP` learn adaptation rates specific to distribution shift?

**Setup**  We evaluate `ATP` on a variety of models, datasets and distribution shifts. We first evaluate on CIFAR-10(-C) with a standard three-way split [50]: we randomly split the dataset to 300 clients: 240 source clients and 60 target clients. Each source client has 160 training samples and 40 validation samples, while each target client has 200 unlabeled testing samples. We simulate three kinds of distribution shifts: feature shift, label shift, and hybrid shift. For feature shift, we follow [12, 17], randomly apply 15 different kinds of corruptions to the source clients, and 4 new kinds of corruptions to the target clients to test the generalization of `ATP`. For label shift, we use the step partition [5], where each client has 8 minor classes with 5 images per class, and 2 major classes with 80 images per class. For the hybrid shift, we apply both step partition and feature perturbations. To test `ATP` under more challenging domain shifts, we then evaluate `ATP` on two domain generalization datasets: Digits-5 [25] and PACS [21]. We adopt the leave-one-domain-out evaluation protocol [10], i.e., one domain is chosen to construct target clients, and the remaining domains are used to construct source clients. We follow similar data preprocessing in [25], while additionally applying step partition to inject label shift. Each domain is divided into 10 clients, leading to 40/10 source/target clients for Digits-5 and 30/10 source/target clients for PACS. For the experiments above, we use ResNet-18 [11] as a common choice in FL experiments [42, 14, 33]. We also test `ATP` with two different architectures: a five-layer CNN on CIFAR-10(-C) and ResNet-50 on CIFAR-100(-C). Detailed experiment settings are given in Appendix C.1.

### 6.1 RQ1: Can `ATP` handle different distribution shift?

We compare `ATP` with three kinds of baseline TTA methods. For *feature shift* methods, we compare to BN-Adapt [38] and Tent [45] which adjusts the batch normalization layers, SHOT [28] which adjusts the feature extractor, T3A [16] which adjusts the final classifier, and MEMO [52] which uses augmentation to adjust the whole network. For *label shift*, we compare to EM [36] which adjusts the label priori unsupervisedly with expectation-maximization, and BBSE [29] which uses the validation data to construct a confusion matrix to estimate the label priori. Since re-training a model with different label weights for each

Table 1: Accuracy (mean $\pm$ s.d. %) on target clients under various distribution shifts on CIFAR-10

| Method | Feature shift | Label shift | Hybrid shift | Avg. Rank |
|---|---|---|---|---|
| No adaptation | $69.42 \pm 0.13$ | $72.98 \pm 0.24$ | $63.68 \pm 0.24$ | 7.7 |
| BN-Adapt | $73.52 \pm 0.22$ | $54.54 \pm 0.10$ | $50.42 \pm 0.39$ | 7.0 |
| SHOT | $71.76 \pm 0.17$ | $48.13 \pm 0.18$ | $44.68 \pm 0.32$ | 9.3 |
| Tent | $71.76 \pm 0.09$ | $50.13 \pm 0.21$ | $46.05 \pm 0.26$ | 8.3 |
| T3A | $69.53 \pm 0.08$ | $71.70 \pm 0.32$ | $62.17 \pm 0.17$ | 8.0 |
| MEMO | $72.43 \pm 0.22$ | $77.30 \pm 0.15$ | $68.07 \pm 0.28$ | 4.3 |
| EM | $65.18 \pm 0.12$ | $\underline{80.73 \pm 0.18}$ | $69.85 \pm 0.43$ | 5.0 |
| BBSE | $63.98 \pm 0.17$ | $79.30 \pm 0.17$ | $67.96 \pm 0.43$ | 6.7 |
| Surgical | $69.85 \pm 0.22$ | $76.00 \pm 0.17$ | $66.94 \pm 0.43$ | 6.3 |
| ATP-batch | $\underline{73.68 \pm 0.10}$ | $79.90 \pm 0.22$ | $\underline{73.05 \pm 0.35}$ | $\underline{2.3}$ |
| ATP-online | $\mathbf{74.06 \pm 0.18}$ | $\mathbf{81.96 \pm 0.14}$ | $\mathbf{75.37 \pm 0.22}$ | $\mathbf{1.0}$ |

client is not realistic in FL. We use the estimated label distribution to adjust the output of a classifier.

Table 2: Accuracy (mean ± s.d. %) on target clients under hybrid shift on Digits-5 and PACS

| Method | Digits-5 | | | | | PACS | | | |
|---|---|---|---|---|---|---|---|---|---|
| | MNIST | SVHN | USPS | SynthDigits | MNIST-M | Art | Cartoon | Photo | Sketch |
| No adaptation | 95.47 ± 0.22 | 52.28 ± 1.45 | 89.62 ± 0.44 | 79.75 ± 0.69 | 55.62 ± 0.80 | 71.57 ± 1.16 | 74.71 ± 0.70 | 90.25 ± 0.75 | 74.20 ± 0.72 |
| BN-Adapt | 94.90 ± 0.29 | 57.57 ± 0.53 | 89.51 ± 0.39 | 75.34 ± 0.48 | 59.68 ± 0.44 | 73.55 ± 0.51 | 71.54 ± 0.55 | 92.07 ± 0.26 | 70.92 ± 0.53 |
| SHOT | 94.69 ± 0.31 | 57.91 ± 0.23 | 89.55 ± 0.69 | 76.43 ± 0.34 | 60.19 ± 0.69 | 69.32 ± 0.67 | 67.77 ± 0.40 | 86.97 ± 0.60 | 59.40 ± 0.91 |
| Tent | 95.48 ± 0.29 | 60.67 ± 0.49 | 91.65 ± 0.61 | 78.56 ± 0.45 | 62.49 ± 0.73 | 71.59 ± 0.71 | 71.03 ± 0.97 | 88.06 ± 0.24 | 63.15 ± 1.10 |
| T3A | 94.63 ± 0.61 | 49.90 ± 1.10 | 88.46 ± 0.75 | 75.47 ± 1.14 | 51.25 ± 1.55 | 72.15 ± 0.72 | 75.02 ± 0.78 | 91.51 ± 0.62 | 70.14 ± 1.21 |
| MEMO | 95.92 ± 0.19 | 52.85 ± 1.09 | 89.84 ± 0.44 | 80.12 ± 0.90 | 55.48 ± 1.13 | 71.47 ± 1.29 | 75.57 ± 0.98 | 90.65 ± 0.90 | 76.30 ± 0.65 |
| EM | 96.64 ± 0.31 | 57.21 ± 1.65 | 92.29 ± 0.32 | 85.69 ± 0.46 | 62.08 ± 0.60 | 73.96 ± 1.85 | 78.91 ± 0.92 | 92.30 ± 0.92 | 80.82 ± 1.52 |
| BBSE | 94.47 ± 0.58 | 57.26 ± 1.47 | 91.34 ± 0.39 | 85.54 ± 0.46 | 61.59 ± 0.91 | 74.33 ± 1.78 | 78.69 ± 1.00 | 91.82 ± 0.68 | 80.15 ± 1.42 |
| Surgical | 97.35 ± 0.13 | 59.93 ± 2.01 | 94.19 ± 0.40 | 86.06 ± 0.44 | 65.87 ± 0.78 | 74.59 ± 2.69 | 77.48 ± 0.64 | 92.34 ± 0.78 | 80.90 ± 3.42 |
| ATP-batch | 97.81 ± 0.27 | 62.18 ± 1.71 | 95.41 ± 0.26 | 87.91 ± 0.45 | 69.98 ± 1.96 | 82.92 ± 0.96 | **79.64 ± 0.75** | 95.40 ± 0.41 | 82.28 ± 1.57 |
| ATP-online | **97.81 ± 0.23** | **62.64 ± 1.92** | **95.56 ± 0.23** | **88.33 ± 0.47** | **70.78 ± 2.36** | **83.51 ± 0.84** | 79.46 ± 0.77 | **95.52 ± 0.40** | **82.80 ± 1.69** |

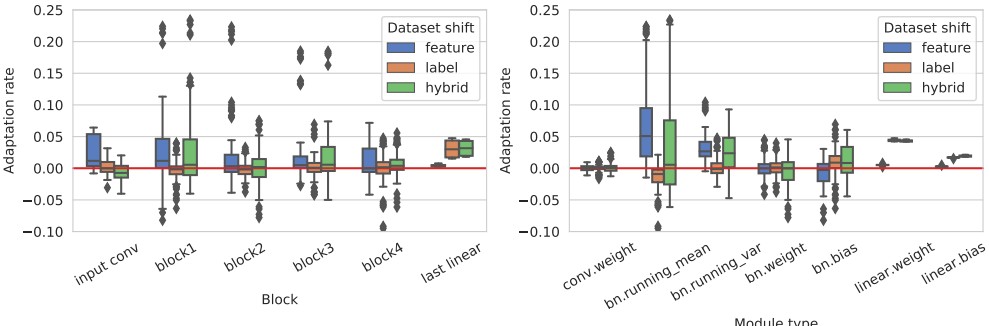

Figure 4: Adaptation rates learned by ATP with different distribution shifts on CIFAR-10

We also compare to Surgical [20] which uses the validation data to decide which blocks to adapt. For all baselines we use the validation data to select hyperparameters.

**ATP can handle different types of distribution shifts**   Table 1 shows the results on CIFAR-10. Under feature and label shifts, most TTA methods suffer from performance trade-off as they improve the performance on one distribution shift while harm the other. The only exception is MEMO, which utilizes data augmentation to robustify the model prediction. However, it also introduces significant computational cost during inference. As an adaptive framework simpler than ours, Surgical also introduces accuracy gain across all distribution shifts. However, its coarse-grained adaptation rule prevents further improvement on the accuracy. ATP reaches great performance comparable to the strongest baseline TTA method under both feature and label shifted. Under the more complex hybrid shift, ATP achieves the highest performance gain with a significant margin. Meanwhile, ATP-online can further improve the performance of ATP-batch by using information from previous batches.

**ATP can handle more challenging domain shifts**   Table 2 shows the results on two domain generalization datasets with a hybrid of domain and label shifts. Compared to baselines, ATP consistently achieves higher accuracy across all domains.

**ATP is compatible to multiple model architectures**   Finally, we evaluate ATP on more model architectures: Shallow-CNN as smaller model and ResNet-50 as larger model. As shown in Table 5 in Appendix C.2, ATP has uniformly good performance on two new models.

### 6.2   RQ2: Does ATP learn adaptation rates specific to distribution shift?

Besides ATP's good performance, we are also interested in whether ATP successfully learns adaptation rates *specific to the type distribution shift*. To explore this, we group the adaptation rates by their corresponding block and module type under three kinds of distribution shifts. As shown in Figure 4, ATP learns significantly different adaptation rates under different distribution shifts. In Figure 4 (left), ATP learns to adapt the last linear layer under label shift, while mainly adapt the former layers under feature shift. More interestingly, we notice in Figure 4 (right) that the adaptation rates for batch norm running statistics are positive under feature shift, but negative under label shift. Negative adaptation rate is usually counter-intuitive, since it disaligns the training and testing distributions. However, it benefits the model under label shifts because it explicitly adapts the label prior distribution towards

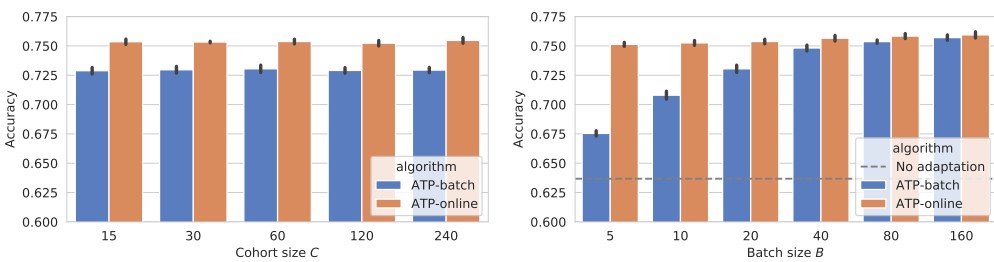

Figure 5: Effect of cohort size and batch size

the prediction distribution. We use a toy example in Appendix C.5 to show why negative adaptation rate can improve the model performance under label distribution.

Moreover, we examine whether the learn adaptation rates are specific to the type of distribution shift by training on one distribution shift, but testing on another. We observe in Table 3 that, ATP performs the best when trained and tested with the same type of distribution shifts. However, the adaptation rates trained on feature/label shift fails to boost the performance on the other distribution shift. The adaptation rates trained on hybrid shift can generalize to feature shift and label shift, but still worse than the adaptation rates trained with the same type of distribution shifts. These results show that the learn adaptation rates are specific to the type of distribution shift.

Table 3: Train and test adaptation rates with different distribution shifts, accuracy (mean $\pm$ s.d. %)

| Train | Test | | |
| | Feature shift | Label shift | Hybrid shift |
| --- | --- | --- | --- |
| No adaptation | 69.42 $\pm$ 0.13 | 72.98 $\pm$ 0.24 | 63.68 $\pm$ 0.24 |
| Feature shift | **73.68 $\pm$ 0.10** | 65.05 $\pm$ 1.82 | 60.64 $\pm$ 1.43 |
| Label shift | 67.99 $\pm$ 0.28 | **79.90 $\pm$ 0.22** | 69.50 $\pm$ 0.52 |
| Hybrid shift | 72.69 $\pm$ 0.14 | 78.92 $\pm$ 0.34 | **73.05 $\pm$ 0.35** |

### 6.3 Further discussion

**Ablation study** We present two variants of ATP to study how trainable parameters and running statistics contribute to the adaptability of ATP. ATP-params only learns to adapt the trainable parameters, while ATP-stats focuses solely on adapting the running statistics. As shown in Table 4, adapting trainable parameters and running statistics both play critical roles in achieving successful adaptation. More specifically, ATP-params primarily facilitate adaptation to label shift, whereas ATP-stats essentially aid in adapting to feature shift.

Table 4: Ablation study, accuracy (mean $\pm$ s.d. %)

| Method | Feature shift | Label shift | Hybrid shift |
| --- | --- | --- | --- |
| No adaptation | 69.42 $\pm$ 0.13 | 72.98 $\pm$ 0.24 | 63.68 $\pm$ 0.24 |
| ATP-params | 69.23 $\pm$ 0.27 | 78.29 $\pm$ 0.14 | 68.05 $\pm$ 0.54 |
| ATP-stats | 71.27 $\pm$ 0.17 | 74.03 $\pm$ 0.18 | 64.78 $\pm$ 0.27 |
| ATP-batch | **73.71 $\pm$ 0.14** | **79.90 $\pm$ 0.22** | **73.05 $\pm$ 0.35** |

**Hyperparameter sensitivity** Figure 5 shows the effects of cohort size and batch size with CIFAR-10 under the hybrid shift, where cohort size refers to the number of clients sampled at each round. ATP demonstrates remarkable consistency in accuracy across different cohort sizes, indicating its robustness. For batch size, we optimize the adaptation rates with $B = 20$ and subsequently evaluate the algorithm with different batch sizes. We find that ATP consistently improves the model's accuracy across different batch sizes, with larger batch sizes yielding greater benefits for the model. ATP-online is more robust to batch size than ATP-batch since it can utilizes information from previous batches.

## 7 Conclusion

In this paper, we propose ATP that unsupervisedly learns the adaptation rate for each module to handle various types of distribution shifts encountered in test-time personalized federated learning. As a potential future direction, incorporating the training of the global model could offer advantages in terms of facilitating easier and better personalization.

## Acknowledgments and Disclosure of Funding

This work is supported by National Science Foundation under Award No. IIS-1947203, IIS-2117902, IIS-2137468, IIS-2002540, Agriculture and Food Research Initiative (AFRI) grant no. 2020-67021-32799/project accession no.1024178 from the USDA National Institute of Food and Agriculture, the U.S. Department of Homeland Security under Grant Award Number, 17STQAC00001-06-00, and IBM-Illinois Discovery Accelerator Institute - a new model of an academic-industry partnership designed to increase access to technology education and skill development to spur breakthroughs in emerging areas of technology. The views and conclusions are those of the authors and should not be interpreted as representing the official policies of the funding agencies or the government.

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

# A  More discussions

## A.1  More related works

**Partial fine-tuning**, i.e., updating a subset of modules of a pretrained network on a new dataset, has been studied in supervised settings [26, 35, 40]. In FL, PartialFed [42] adaptively decides whether each parameter is shared or personalized. However, it cannot generalize to testing clients that do not participate in the training. Recently, surgical fine-tuning [20] selectively fine-tunes a subset of blocks with a similar intuition that the type of distribution shift influences which part of the network to be adapted. Different from their method, we focus on the unsupervised setting and propose to refine the adaptation rate for each module.

**Hyperparameter optimization** is also related to our algorithm if considering adaptation rates as a set of hyperparameters. [19] first investigates the problem of federated hyperparameter tuning and proposed FedEX that leverages weight-sharing from neural architecture search to efficiently tune hyperparameters. [55] introduces FloRA that addresses use cases of tabular data and enables single-shot federated hyperparameter tuning. While these methods focus on improving the efficiency of hyperparameter optimization, our paper focuses on finding the optimal adaptation rates that benefit test-time personalization.

## A.2  Broader impacts and limitations

**Broader impacts**  We are not aware of any potential negative societal impacts regarding our work to the best of our knowledge. For all the used data sets, there is no private personally identifiable information or offensive content.

**Limitations**  One possible limitation is that we consider a fix global model for lower communication cost and better generalization, while it might be beneficial to also train a global model for easier personalization, which could be a promising future direction.

# B   Theoretical analysis

In this section, we give theoretical proofs of convergence, and generalization of `ATP`.

## B.1   Approximation analysis

In this subsection, we give detailed proofs of Proposition 3.1 and 3.2 in Section 3 of the main text. These propositions show why certain types of distribution shifts can be handled by adapting certain layers in a neural network.

### B.1.1   Proof of Proposition 3.1

**Proposition 3.1** (Adapting the last layer to handle label shift). Consider two distribution $p, q$ with $p(\boldsymbol{x}|\boldsymbol{y}) = q(\boldsymbol{x}|\boldsymbol{y})$ and $p(\boldsymbol{y}) \neq q(\boldsymbol{y})$. When a neural network is calibrated on $p$, i.e., $f(\boldsymbol{x}; \boldsymbol{w}) = p(\cdot|\boldsymbol{x})$, it is calibrated on $q$ after adding $\log \frac{q(\boldsymbol{y})}{p(\boldsymbol{y})}$ to the bias term of the final last layer.

*Proof.* W.l.o.g., assuming the last layer of the neural network is a linear layer. Denoting $g(\boldsymbol{x}; \boldsymbol{w}_g)$ as the input of the last layer, where $\boldsymbol{x}$ is the input and $\boldsymbol{w}_g$ is the model parameters for the feature extractor (i.e., all layers except for the last classification layer). Denote $\boldsymbol{w}_1, \cdots, \boldsymbol{w}_K$ as the weights of the last layer and $b_1, \cdots, b_K$ as the bias terms of the last layer, assuming $K$ classes. Then we have

$$f(\boldsymbol{x}; \boldsymbol{w})_c = \frac{\exp(\boldsymbol{w}_c^\top g(\boldsymbol{x}; \boldsymbol{w}_g) + b_c)}{\sum_{c'=1}^{K} \exp(\boldsymbol{w}_{c'}^\top g(\boldsymbol{x}; \boldsymbol{w}_g) + b_{c'})}$$

Since the neural network is calibrated on $p$, for all class index $c = 1, \cdots, K$, we have

$$f(\boldsymbol{x}, \boldsymbol{w})_c = p(\boldsymbol{y} = \boldsymbol{e}_c|\boldsymbol{x})$$

where $\boldsymbol{e}_c$ is an one-hot vector with its $c$-th element as one. For distribution $q$ with the same conditional distribution and different priori, by Bayes' theorem, $\forall \boldsymbol{x}, \boldsymbol{y}$

$$q(\boldsymbol{y}|\boldsymbol{x}) = \frac{q(\boldsymbol{x}|\boldsymbol{y})q(\boldsymbol{y})}{\sum_{\boldsymbol{y}} q(\boldsymbol{x}|\boldsymbol{y})q(\boldsymbol{y})} = \frac{p(\boldsymbol{x}|\boldsymbol{y})q(\boldsymbol{y})}{\sum_{\boldsymbol{y}} p(\boldsymbol{x}|\boldsymbol{y})q(\boldsymbol{y})} = \frac{p(\boldsymbol{y}|\boldsymbol{x}) \cdot \frac{q(\boldsymbol{y})}{p(\boldsymbol{y})}}{\sum_{\boldsymbol{y}} p(\boldsymbol{y}|\boldsymbol{x}) \cdot \frac{q(\boldsymbol{y})}{p(\boldsymbol{y})}}$$

Therefore, we can calibrate the neural network on distribution $q$ simply by adding $\log \frac{q(\boldsymbol{y})}{p(\boldsymbol{y})}$ to the bias terms, i.e.,

$$
\begin{aligned}
f_{cal}(\boldsymbol{x}; \boldsymbol{w}_{cal})_c &= \frac{\exp(\boldsymbol{w}_c^\top g(\boldsymbol{x}; \boldsymbol{w}_g) + b_c + \log \frac{q(\boldsymbol{e}_c)}{p(\boldsymbol{e}_c)})}{\sum_{c'=1}^{K} \exp(\boldsymbol{w}_{c'}^\top g(\boldsymbol{x}; \boldsymbol{w}_g) + b_{c'} + \log \frac{q(\boldsymbol{e}_{c'})}{p(\boldsymbol{e}_{c'})})} \\
&= \frac{\exp(\boldsymbol{w}_c^\top g(\boldsymbol{x}; \boldsymbol{w}_g) + b_c) \cdot \frac{q(\boldsymbol{e}_c)}{p(\boldsymbol{e}_c)}}{\sum_{c'=1}^{K} \exp(\boldsymbol{w}_{c'}^\top g(\boldsymbol{x}; \boldsymbol{w}_g) + b_{c'}) \cdot \frac{q(\boldsymbol{e}_{c'})}{p(\boldsymbol{e}_{c'})}} \\
&= \frac{p(\boldsymbol{e}_c|\boldsymbol{x}) \cdot \frac{q(\boldsymbol{e}_c)}{p(\boldsymbol{e}_c)}}{\sum_{c'=1}^{K} p(\boldsymbol{e}_{c'}|\boldsymbol{x}) \cdot \frac{q(\boldsymbol{e}_{c'})}{p(\boldsymbol{e}_{c'})}} \\
&= q(\boldsymbol{y} = \boldsymbol{e}_c|\boldsymbol{x})
\end{aligned}
$$

$\square$

### B.1.2 Proof of Proposition 3.2

**Proposition 3.2** (Adapting the BN layer to handle feature shift [38])**.** When the feature shift only causes differences in the first and second order moments of the feature activations $z = g(x)$ where $g$ is the combination of layers before the BN layer, assuming independent activations, the feature shift can be removed by adapting running mean and variance of the BN layer.

*Proof.* Denote the source and target feature (marginal) distributions to be $p(x)$ and $q(x)$. Given independent, activations, we only need to test the marginal distribution of each $z \in z = g(x)$. For each $z$, since the feature shift only introduces differences in the first and second order moments, there exists $\Delta$ and $r > 0$, s.t., $\forall z_t \in \mathbb{R}$

$$\Pr_{x \sim q} (z \geq z_t) = \Pr_{x \sim p} \left( z \geq \frac{z_t - \Delta}{r} \right)$$

which indicates that the distribution of $z$ is first shifted by $\Delta$ and then scaled by $r$. Such distribution shift in the feature activation can be removed by adapting the running mean $\mu_p$ and variance $\sigma_p^2$

$$\mu_q = r \cdot \mu_p + \Delta$$
$$\sigma_q = \sigma_p \cdot r$$

As a result, for all $t \in \mathbb{R}$

$$
\begin{aligned}
\Pr_{x \sim q} \left( \frac{z - \mu_q}{\sigma_q} \geq t \right) &= \Pr_{x \sim q} (z \geq \mu_q + \sigma_q \cdot t) \\
&= \Pr_{x \sim p} \left( z \geq \frac{\mu_q + \sigma_q \cdot t - \Delta}{r} \right) \\
&= \Pr_{x \sim p} (z \geq \mu_p + \sigma_p \cdot t) \\
&= \Pr_{x \sim p} \left( \frac{z - \mu_p}{\sigma_p} \geq t \right)
\end{aligned}
$$

which indicates that the feature shift is removed after normalization with running statistics $\mu_q, \sigma_q$. $\square$

## B.2 Convergence analysis

In this part, we show that `ATP` has the same convergence guarantee as FedAvg [31]. We first show in Lemma B.5 and B.10 that `ATP` preserves convexity and smoothness, which are two important conditions in the analysis of convergence. Then we formally prove the convergence of `ATP` in Theorem B.11.

### B.2.1 Definitions: local and global objective

For clarity, we first formally define the data generation process, and local/global objectives for optimization.

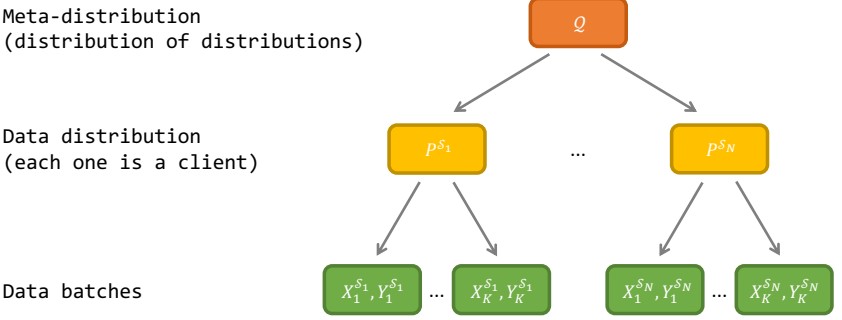

Figure 6: Data generation process

**Data generation** We consider a two-stage sampling process as illustrated in Figure 6.

- There are $N$ source clients' distributions $P^{\mathcal{S}_1}, P^{\mathcal{S}_2}, \cdots, P^{\mathcal{S}_N}$ and $M$ target clients' distribution $P^{\mathcal{T}_1}, P^{\mathcal{T}_2}, \cdots, P^{\mathcal{T}_M}$ i.i.d. drawn from a meta-distribution $\mathcal{Q}$.
- For each source client $i$'s distribution $P^{\mathcal{S}_i}$, there are $K$ **data batches** $(\boldsymbol{X}_1^{\mathcal{S}_i}, \boldsymbol{Y}_1^{\mathcal{S}_i})$, $(\boldsymbol{X}_2^{\mathcal{S}_i}, \boldsymbol{Y}_2^{\mathcal{S}_i}), \cdots, (\boldsymbol{X}_K^{\mathcal{S}_i}, \boldsymbol{Y}_K^{\mathcal{S}_i})$ drawn i.i.d. from $P^{\mathcal{S}_i}$.
- Each batch consists of $B$ samples, $(\boldsymbol{X}_k^{\mathcal{S}_i}, \boldsymbol{Y}_k^{\mathcal{S}_i}) = \{(\boldsymbol{x}_{k,b}^{\mathcal{S}_i}, \boldsymbol{y}_{k,b}^{\mathcal{S}_i})\}_{b=1}^B$ where $B$ is the batch size.
- For simplicity, we assume that all source client has the same number of batches $K$ and batch size $B$.

**Definition B.1** (Batch objective). Define the batch objective of the $k$-th batch on client $\mathcal{S}_i$ to be

$$F_{ik}(\boldsymbol{\alpha}) = \frac{1}{B} \sum_{b=1}^B \ell_{CE}(f(\boldsymbol{x}_{k,b}^{\mathcal{S}_i}, \boldsymbol{w}_k^{\mathcal{S}_i}, \boldsymbol{y}_{k,b}^{\mathcal{S}_i})$$

where $\boldsymbol{w}_k^{\mathcal{S}_i} = \boldsymbol{w}_G + (\boldsymbol{A}\boldsymbol{\alpha}) \odot \boldsymbol{h}_k^{\mathcal{S}_i}$ and $\boldsymbol{h}_k^{\mathcal{S}_i}$ is the update direction computed with $\boldsymbol{X}_k^{\mathcal{S}_i} = \{\boldsymbol{x}_{k,b}^{\mathcal{S}_i}\}_{b=1}^B$ with Eq. (4) and (5).

**Definition B.2** (Local objective). Define the local objective of client $i$ to be

$$F_i(\boldsymbol{\alpha}) = \frac{1}{K} \sum_{k=1}^K F_{ik}(\boldsymbol{\alpha})$$

**Definition B.3** (Global objective). Define the global objective to be

$$F(\boldsymbol{\alpha}) = \frac{1}{N} \sum_{i=1}^N F_i(\boldsymbol{\alpha})$$

### B.2.2 `ATP` preserves convexity and smoothness

In this part, we show that `ATP` preserves convexity and smoothness, which are two important conditions in the analysis of convergence.

**Definition B.4** (Convexity). A function $f : \mathbb{R}^D \to \mathbb{R}$ is convex if for all $\boldsymbol{x}_1, \boldsymbol{x}_2 \in \mathbb{R}^D$ and $\lambda \in [0, 1]$

$$f(\lambda \boldsymbol{x}_1 + (1 - \lambda)\boldsymbol{x}_2) \leq \lambda f(\boldsymbol{x}_1) + (1 - \lambda)f(\boldsymbol{x}_2)$$

**Lemma B.5** (Convexity preserving). *If $\ell_{CE}(f(\boldsymbol{x}; \boldsymbol{w}), \boldsymbol{y})$ is convex w.r.t. $\boldsymbol{w}$ given any data sample $(\boldsymbol{x}, \boldsymbol{y})$, then $F_i(\boldsymbol{\alpha})$ is convex w.r.t. $\boldsymbol{\alpha}$.*

*Proof.* Noticing that $\boldsymbol{w}_k^{\mathcal{S}_i} = \boldsymbol{w}_G + (\boldsymbol{A}\boldsymbol{\alpha}) \odot \boldsymbol{h}_k^{\mathcal{S}_i}$ is linear to $\boldsymbol{\alpha}$, linear transformation preserves convexity. For any update direction $\boldsymbol{h}_k^{\mathcal{S}_i}$ and data sample $(\boldsymbol{x}_{k,b}^{\mathcal{S}_i}, \boldsymbol{y}_{k,b}^{\mathcal{S}_i})$, we find that

$$\ell_{CE}(f(\boldsymbol{x}_{k,b}^{\mathcal{S}_i}; \boldsymbol{w}_G + (\boldsymbol{A}(\lambda\boldsymbol{\alpha}_1 + (1-\lambda)\boldsymbol{\alpha}_2)) \odot \boldsymbol{h}_k^{\mathcal{S}_i}, \boldsymbol{y}_{k,b}^{\mathcal{S}_i})$$

$$= \ell_{CE}(f(\boldsymbol{x}_{k,b}^{\mathcal{S}_i}; \lambda\left[\boldsymbol{w}_G + (\boldsymbol{A}\boldsymbol{\alpha}_1) \odot \boldsymbol{h}_k^{\mathcal{S}_i}\right] + (1-\lambda)\left[\boldsymbol{w}_G + (\boldsymbol{A}\boldsymbol{\alpha}_2) \odot \boldsymbol{h}_k^{\mathcal{S}_i}\right], \boldsymbol{y}_{k,b}^{\mathcal{S}_i})$$

$$\leq \lambda\ell_{CE}(f(\boldsymbol{x}_{k,b}^{\mathcal{S}_i}; \boldsymbol{w}_G + (\boldsymbol{A}\boldsymbol{\alpha}_1) \odot \boldsymbol{h}_k^{\mathcal{S}_i}, \boldsymbol{y}_{k,b}^{\mathcal{S}_i}) + (1-\lambda)\ell_{CE}(f(\boldsymbol{x}_{k,b}^{\mathcal{S}_i}; \boldsymbol{w}_G + (\boldsymbol{A}\boldsymbol{\alpha}_2) \odot \boldsymbol{h}_k^{\mathcal{S}_i}, \boldsymbol{y}_{k,b}^{\mathcal{S}_i})$$

i.e., $\ell_{CE}(f(\boldsymbol{x}_{k,b}^{\mathcal{S}_i}; \boldsymbol{w}_G + (\boldsymbol{A}\boldsymbol{\alpha}) \odot \boldsymbol{h}_k^{\mathcal{S}_i}), \boldsymbol{y}_{k,b}^{\mathcal{S}_i})$ is convex w.r.t. $\boldsymbol{\alpha}$.

Finally, since

$$F_i(\boldsymbol{\alpha}) = \frac{1}{KB}\sum_{k=1}^{K}\sum_{b=1}^{B}\ell_{CE}(f(\boldsymbol{x}_{k,b}^{\mathcal{S}_i}; \boldsymbol{w}_G + (\boldsymbol{A}\boldsymbol{\alpha}) \odot \boldsymbol{h}_k^{\mathcal{S}_i}), \boldsymbol{y}_{k,b}^{\mathcal{S}_i})$$

which is the average of $KB$ convex functions, we have that $F_i(\boldsymbol{\alpha})$ is also convex to $\boldsymbol{\alpha}$. $\square$

**Definition B.6** ($\beta$-smoothness). A function $f : \mathbb{R}^D \to \mathbb{R}$ is $L$-smoothness with $\beta > 0$ if for all $\boldsymbol{x}_1, \boldsymbol{x}_2 \in \mathbb{R}^D$,

$$\|\nabla f(\boldsymbol{x}_1) - \nabla f(\boldsymbol{x}_2)\|_2 \leq \beta\|\boldsymbol{x}_1 - \boldsymbol{x}_2\|_2$$

**Definition B.7** ($H$-module-wise-bounded update direction). The update direction is $H$-module-wise-bounded for a data batch $\boldsymbol{X}_k^{\mathcal{S}_i}$ if

$$\|(\boldsymbol{h}_k^{\mathcal{S}_i})^{[l]}\|_2 \leq H, \quad \forall l = 1, \cdots, d$$

where $(\boldsymbol{h}_k^{\mathcal{S}_i})^{[l]}$ is the update direction corresonding to the $l$-th module and $d$ is the number of modules in the neural network.

**Lemma B.8** (Lipschitz parameter). *If the update direction is $H$-module-wise-bounded for a data batch $\boldsymbol{X}_k^{\mathcal{S}_i}$. Given two adaptation rates $\boldsymbol{\alpha}_1, \boldsymbol{\alpha}_2$ and the global model $\boldsymbol{w}_G$, we have*

$$\|\boldsymbol{w}_k^{\mathcal{S}_i}(\boldsymbol{\alpha}_1) - \boldsymbol{w}_k^{\mathcal{S}_i}(\boldsymbol{\alpha}_2)\|_2 \leq H \cdot \|\boldsymbol{\alpha}_1 - \boldsymbol{\alpha}_2\|_2$$

*where $\boldsymbol{w}_k^{\mathcal{S}_i}(\boldsymbol{\alpha}_1) = \boldsymbol{w}_G + (\boldsymbol{A}\boldsymbol{\alpha}_1) \odot \boldsymbol{h}_k^{\mathcal{S}_i}$ is the personalized model updated with $\boldsymbol{\alpha}_1$ as the adaptation rate.*

*Proof.*

$$\|\boldsymbol{w}_k^{\mathcal{S}_i}(\boldsymbol{\alpha}_1) - \boldsymbol{w}_k^{\mathcal{S}_i}(\boldsymbol{\alpha}_2)\|_2 = \|(\boldsymbol{w}_G + (\boldsymbol{A}\boldsymbol{\alpha}_1) \odot \boldsymbol{h}_k^{\mathcal{S}_i}) - (\boldsymbol{w}_G + (\boldsymbol{A}\boldsymbol{\alpha}_2) \odot \boldsymbol{h}_k^{\mathcal{S}_i})\|_2$$

$$= \|(\boldsymbol{A}(\boldsymbol{\alpha}_1 - \boldsymbol{\alpha}_2)) \odot \boldsymbol{h}_k^{\mathcal{S}_i}\|_2$$

$$= \|\boldsymbol{h}_k^{\mathcal{S}_i} \odot (\boldsymbol{A}(\boldsymbol{\alpha}_1 - \boldsymbol{\alpha}_2))\|_2$$

$$= \sqrt{\sum_{l=1}^{d}\left\|(\boldsymbol{h}_k^{\mathcal{S}_i})^{[l]}\right\|_2^2\left(\alpha_1^{[l]} - \alpha_2^{[l]}\right)^2}$$

$$\leq \sqrt{\sum_{l=1}^{d}H^2\left(\alpha_1^{[l]} - \alpha_2^{[l]}\right)^2}$$

$$= H \cdot \|\boldsymbol{\alpha}_1 - \boldsymbol{\alpha}_2\|_2$$

$\square$

*Remark* B.9. Lemma B.8 indicates that when the adaptation rate is perturbed by a little, the personalized model parameter $\boldsymbol{w}_k^{\mathcal{S}_i}$ is also only perturbed by a little.

**Lemma B.10** (Smoothness preserving). *If (1) $\ell_{CE}(f(\boldsymbol{x};\boldsymbol{w}),\boldsymbol{y})$ is $\beta$-smooth w.r.t. $\boldsymbol{w}$ given any data sample $(\boldsymbol{x},\boldsymbol{y})$, and (2) the update direction $\boldsymbol{h}_k^{\mathcal{S}_i}$ is $H$-module-wise-bounded for all data batches $\boldsymbol{X}_k^{\mathcal{S}_i}$, then $F_i(\boldsymbol{\alpha})$ is $(H^2\beta)$-smoothness w.r.t. $\boldsymbol{\alpha}$.*

*Proof.* We first give an upper bound of $\|\boldsymbol{A}^\top\mathrm{diag}(\boldsymbol{h}_k^{\mathcal{S}_i})\|_2$ when $\boldsymbol{h}_k^{\mathcal{S}_i}$ is $H$-module-wise-bounded. The update direction $\boldsymbol{h}_k^{\mathcal{S}_i}\in\mathbb{R}^D$ is the concatenation of update directions for each module $\{(\boldsymbol{h}_k^{\mathcal{S}_i})^{[l]}\}_{l=1}^d$, i.e.,

$$\left(\boldsymbol{h}_k^{\mathcal{S}_i}\right)^\top = \left[\left((\boldsymbol{h}_k^{\mathcal{S}_i})^{[1]}\right)^\top, \cdots, \left((\boldsymbol{h}_k^{\mathcal{S}_i})^{[d]}\right)^\top\right]$$

where $(\boldsymbol{h}_k^{\mathcal{S}_i})^{[l]}$ is a column vector representing the update direction of the $l$-th module in the model. Similarly, any other vector $\boldsymbol{v}\in\mathbb{R}^D$ can be correspondingly expressed as

$$\boldsymbol{v}^\top = \left[\left(\boldsymbol{v}^{[1]}\right)^\top, \cdots, \left(\boldsymbol{v}^{[d]}\right)^\top\right]$$

Then,

$$
\begin{aligned}
\|\boldsymbol{A}^\top\mathrm{diag}(\boldsymbol{h}_k^{\mathcal{S}_i})\|_2 &= \sup_{\boldsymbol{v}\in\mathbb{R}^D}\frac{\|\boldsymbol{A}^\top\mathrm{diag}(\boldsymbol{h}_k^{\mathcal{S}_i})\boldsymbol{v}\|_2}{\|\boldsymbol{v}\|_2}\\
&= \sup_{\boldsymbol{v}\in\mathbb{R}^D}\frac{\|\boldsymbol{A}^\top(\boldsymbol{h}_k^{\mathcal{S}_i}\odot\boldsymbol{v})\|_2}{\|\boldsymbol{v}\|_2}\\
&= \sup_{\boldsymbol{v}\in\mathbb{R}^D}\sqrt{\frac{\sum_{l=1}^d\left[\left((\boldsymbol{h}_k^{\mathcal{S}_i})^{[l]}\right)^\top\boldsymbol{v}^{[l]}\right]^2}{\sum_{l=1}^d\left\|\boldsymbol{v}^{[l]}\right\|_2^2}}\\
&\leq \sup_{\boldsymbol{v}\in\mathbb{R}^D}\sqrt{\frac{\sum_{l=1}^d\left[\left\|(\boldsymbol{h}_k^{\mathcal{S}_i})^{[l]}\right\|_2\cdot\left\|\boldsymbol{v}^{[l]}\right\|_2\right]^2}{\sum_{l=1}^d\left\|\boldsymbol{v}^{[l]}\right\|_2^2}}\\
&\leq \sup_{\boldsymbol{v}\in\mathbb{R}^D}\sqrt{\frac{\sum_{l=1}^d\left[H\cdot\left\|\boldsymbol{v}^{[l]}\right\|_2\right]^2}{\sum_{l=1}^d\left\|\boldsymbol{v}^{[l]}\right\|_2^2}} && \text{(Definition B.7)}\\
&= H
\end{aligned}
$$

We then prove that for any $H$-module-wise-bounded update direction $\boldsymbol{h}_k^{\mathcal{S}_i}$ and data sample $(\boldsymbol{x}_{k,b}^{\mathcal{S}_i},\boldsymbol{y}_{k,b}^{\mathcal{S}_i})$, we have $\ell_{CE}(f(\boldsymbol{x}_{k,b}^{\mathcal{S}_i};\boldsymbol{w}_k^{\mathcal{S}_i}(\boldsymbol{\alpha})),\boldsymbol{y}_{k,b}^{\mathcal{S}_i})$ is $H^2\beta$-smoothness w.r.t. $\boldsymbol{\alpha}$.

$$
\begin{aligned}
&\|\nabla_{\boldsymbol{\alpha}_1}\ell_{CE}(f(\boldsymbol{x}_{k,b}^{\mathcal{S}_i};\boldsymbol{w}_k^{\mathcal{S}_i}(\boldsymbol{\alpha}_1)),\boldsymbol{y}_{k,b}^{\mathcal{S}_i}) - \nabla_{\boldsymbol{\alpha}_2}\ell_{CE}(f(\boldsymbol{x}_{k,b}^{\mathcal{S}_i};\boldsymbol{w}_k^{\mathcal{S}_i}(\boldsymbol{\alpha}_2)),\boldsymbol{y}_{k,b}^{\mathcal{S}_i})\|_2\\
&= \|\boldsymbol{A}^\top(\boldsymbol{h}_k^{\mathcal{S}_i}\odot\nabla_{\boldsymbol{w}_k^{\mathcal{S}_i}(\boldsymbol{\alpha}_1)}\ell_{CE}(f(\boldsymbol{x}_{k,b}^{\mathcal{S}_i};\boldsymbol{w}_k^{\mathcal{S}_i}(\boldsymbol{\alpha}_1)),\boldsymbol{y}_{k,b}^{\mathcal{S}_i}) - \boldsymbol{A}^\top(\boldsymbol{h}_k^{\mathcal{S}_i}\odot\nabla_{\boldsymbol{w}_k^{\mathcal{S}_i}(\boldsymbol{\alpha}_2)}\ell_{CE}(f(\boldsymbol{x}_{k,b}^{\mathcal{S}_i};\boldsymbol{w}_k^{\mathcal{S}_i}(\boldsymbol{\alpha}_2)),\boldsymbol{y}_{k,b}^{\mathcal{S}_i})\|_2\\
&\leq \|\boldsymbol{A}^\top\mathrm{diag}(\boldsymbol{h}_k^{\mathcal{S}_i})\|_2\cdot\|\nabla_{\boldsymbol{w}_k^{\mathcal{S}_i}(\boldsymbol{\alpha}_1)}\ell_{CE}(f(\boldsymbol{x}_{k,b}^{\mathcal{S}_i};\boldsymbol{w}_k^{\mathcal{S}_i}(\boldsymbol{\alpha}_1)),\boldsymbol{y}_{k,b}^{\mathcal{S}_i}) - \nabla_{\boldsymbol{w}_k^{\mathcal{S}_i}(\boldsymbol{\alpha}_2)}\ell_{CE}(f(\boldsymbol{x}_{k,b}^{\mathcal{S}_i};\boldsymbol{w}_k^{\mathcal{S}_i}(\boldsymbol{\alpha}_2)),\boldsymbol{y}_{k,b}^{\mathcal{S}_i})\|_2\\
&\leq \|\boldsymbol{A}^\top\mathrm{diag}(\boldsymbol{h}_k^{\mathcal{S}_i})\|_2\cdot\beta\cdot\|\boldsymbol{w}_k^{\mathcal{S}_i}(\boldsymbol{\alpha}_1) - \boldsymbol{w}_k^{\mathcal{S}_i}(\boldsymbol{\alpha}_2)\|_2 && \text{(Definition B.6)}\\
&\leq \|\boldsymbol{A}^\top\mathrm{diag}(\boldsymbol{h}_k^{\mathcal{S}_i})\|_2\cdot\beta\cdot H\cdot\|\boldsymbol{\alpha}_1 - \boldsymbol{\alpha}_2\|_2 && \text{(Lemma B.8)}\\
&\leq H^2\cdot\beta\cdot\|\boldsymbol{\alpha}_1 - \boldsymbol{\alpha}_2\|_2
\end{aligned}
$$

$\square$

### B.2.3 Convergence of `ATP` under FedAvg framework

Finally, we show that with preservation of convexity and smoothness, `ATP` shares the same convergence guarantee as FedAvg [31]. We apply the proof in [46].

**Theorem B.11** (Convergence of `ATP`). *Assume that*

1. *At any round $t$, each client takes $\tau$ SGD steps with learning rate $\eta$.*

2. *Full participation, i.e., each source client participates every round*

3. *$\ell_{CE}(f(\boldsymbol{x}; \boldsymbol{w}), \boldsymbol{y})$ is convex and $\beta$-smooth w.r.t. $\boldsymbol{w}$ given any data sample $(\boldsymbol{x}, \boldsymbol{y})$.*

4. *The update direction $\boldsymbol{h}_k^{S_i}$ is $H$-module-wise-bounded for all $i, j$*

5. *Bounded inner variance: for any $\boldsymbol{\alpha}$ and client $i$,*

$$\mathbb{E}_j \nabla_{\boldsymbol{\alpha}} F_{ij}(\boldsymbol{\alpha}) = \nabla_{\boldsymbol{\alpha}} F_i(\boldsymbol{\alpha}), \quad \mathbb{E}_j \|\nabla_{\boldsymbol{\alpha}} F_{ij}(\boldsymbol{\alpha}) - \nabla_{\boldsymbol{\alpha}} F_i(\boldsymbol{\alpha})\|_2^2 \le \sigma^2$$

6. *Bounded outer variance: for any $\boldsymbol{\alpha}$ and client $i$,*

$$\|\nabla_{\boldsymbol{\alpha}} F_i(\boldsymbol{\alpha}) - \nabla_{\boldsymbol{\alpha}} F(\boldsymbol{\alpha})\|_2^2 \le \zeta^2$$

*If the client learning rate satisfies $\eta \ge \frac{1}{4H^2\beta}$, then one has*

$$\mathbb{E}\left[\frac{1}{\tau T} \sum_{t=0}^{T-1} \sum_{k=1}^{\tau} F(\bar{\boldsymbol{\alpha}}^{t,k}) - F(\boldsymbol{\alpha}^*)\right] \le \frac{\|\boldsymbol{\alpha}_G^0 - \boldsymbol{\alpha}^*\|_2^2}{2\eta\tau T} + \frac{\eta\sigma^2}{N} + 4\tau\eta^2 H^2 \beta\sigma^2 + 18\tau^2\eta^2 H^2 \beta\zeta^2$$

*where $\boldsymbol{\alpha}^* = \arg\min_{\boldsymbol{\alpha}} F(\boldsymbol{\alpha})$ and $\bar{\boldsymbol{\alpha}}^{t,k} = \frac{1}{N} \sum_{i=1}^{N} \boldsymbol{\alpha}_i^{t,k}$. $\boldsymbol{\alpha}_i^{t,k}$ is the local adaptation rates after $t$ communication rounds and $k$ local epochs.*

*Proof.* The optimization process of `ATP` is similar as FedAvg [31], where the difference is that `ATP` adapts the adaptation rates instead of model parameter. Lemma B.5 and B.10 that `ATP` preserves convexity and smoothness, i.e., for each client $i$, $F_i(\boldsymbol{\alpha})$ is convex and $(H^2\beta)$-smoothness. Therefore, we can apply Theorem 1 in [46] to complete the proof. □

*Remark* B.12. The convergence rate of `ATP` is $\mathcal{O}(\frac{1}{\tau T})$.

## B.3 Generalization analysis

In this part, we studied how an adaptation rate $\boldsymbol{\alpha}$ learned by ATP that performs well on source clients can generalize to target clients. More specifically, we are interested in *how many different source clients are required to ensure a certain generalization error*. Similar to most of the other generalization analysis, we (1) derive generalization bound for any fixed hypothesis ($\boldsymbol{\alpha}$), and (2) quantify the size of hypothesis space.

### B.3.1 Definitions: data generation and error rates

We first formally define the error rates.

**Definition B.13** (Error rate for one data sample). Let $f(\cdot; \boldsymbol{w}_k^{\mathcal{S}_i}) : \mathcal{X} \to \Delta^{|\mathcal{Y}|-1}$ be the neural network with model parameters $\boldsymbol{w}_k^{\mathcal{S}_i}$ that takes *one* data sample $\boldsymbol{x}_{k,b}^{\mathcal{S}_i}$ as input and outputs a probability distribution over the label space, i.e., $f(\boldsymbol{x}_{k,b}^{\mathcal{S}_i}; \boldsymbol{w}_k^{\mathcal{S}_i}) \geq \boldsymbol{0}$ and $\boldsymbol{1}^\top f(\boldsymbol{x}_{k,b}^{\mathcal{S}_i}; \boldsymbol{w}_k^{\mathcal{S}_i}) = 1$. Given adapted model parameters $\boldsymbol{w}_k^{\mathcal{S}_i}$, define the error rate on one data sample $(\boldsymbol{x}_{k,b}^{\mathcal{S}_i}, \boldsymbol{y}_{k,b}^{\mathcal{S}_i})$ to be

$$\hat{e}_{ikb}(\boldsymbol{w}_k^{\mathcal{S}_i}) := 1 - (\boldsymbol{y}_{k,b}^{\mathcal{S}_i})^\top f(\boldsymbol{x}_{k,b}^{\mathcal{S}_i}; \boldsymbol{w}_k^{\mathcal{S}_i})$$

*Remark* B.14. Definition B.13 is equivalent to the expected misclassification rate if when making random decision based on the output probability $f(\boldsymbol{x}_{k,b}^{\mathcal{S}_i}; \boldsymbol{w}_k^{\mathcal{S}_i})$.

**Definition B.15** (Error rate for one data batch). Given global model parameter $\boldsymbol{w}_G$, adaptation rate $\boldsymbol{\alpha}$, and a batch of data $\boldsymbol{X}_k^{\mathcal{S}_i} = \{\boldsymbol{x}_{k,b}^{\mathcal{S}_i}\}_{b=1}^B, \boldsymbol{Y}_k^{\mathcal{S}_i} = \{\boldsymbol{y}_{k,b}^{\mathcal{S}_i}\}_{b=1}^B$, define the error rate for one data batch

$$\hat{\varepsilon}_{ik}(\boldsymbol{\alpha}) := \hat{e}_{ik}(\boldsymbol{w}_k^{\mathcal{S}_i}) := \frac{1}{B} \sum_{b=1}^B \hat{e}_{ikb}(\boldsymbol{w}_k^{\mathcal{S}_i})$$

where

$$\boldsymbol{w}_k^{\mathcal{S}_i} = \boldsymbol{w}_G + (\boldsymbol{A}\boldsymbol{\alpha}) \odot \boldsymbol{h}_k^{\mathcal{S}_i}$$

and $\boldsymbol{h}_k^{\mathcal{S}_i}$ is the update direction computed with $\boldsymbol{X}_k^{\mathcal{S}_i}$.

**Definition B.16** (Error rates for one client). Given global model parameter $\boldsymbol{w}_G$, adaptation rate $\boldsymbol{\alpha}$, and a source client $\mathcal{S}_i$ with $K$ data batches $\{(\boldsymbol{X}_k^{\mathcal{S}_i}, \boldsymbol{Y}_k^{\mathcal{S}_i})\}_{k=1}^K$, define the *empirical error rate* for source client $\mathcal{S}_i$

$$\hat{\varepsilon}_i(\boldsymbol{\alpha}) := \frac{1}{K} \sum_{k=1}^K \hat{\varepsilon}_{ik}(\boldsymbol{\alpha})$$

Also, define the *expected error rate* for source client $\mathcal{S}_i$

$$\varepsilon_i(\boldsymbol{\alpha}) := \mathbb{E}_{(\boldsymbol{X}_k^{\mathcal{S}_i}, \boldsymbol{Y}_k^{\mathcal{S}_i}) \sim P^{\mathcal{S}_i}} \left[ \hat{\varepsilon}_{ik}(\boldsymbol{\alpha}) \right]$$

*Remark* B.17. $\hat{\varepsilon}_i(\boldsymbol{\alpha})$ quantifies the error rate on client $\mathcal{S}_i$'s finite dataset $\mathbb{D}^{\mathcal{S}_i} = \{(\boldsymbol{X}_k^{\mathcal{S}_i}, \boldsymbol{Y}_k^{\mathcal{S}_i})\}_{k=1}^K$. $\varepsilon_i(\boldsymbol{\alpha})$ quantifies the expected error rate on a *new data batch* from client $\mathcal{S}_i$. Notice that the same definition applies to target clients.

**Definition B.18** (Source error rate and expected target error rate). Given global model parameter $\boldsymbol{w}_G$, adaptation rate $\boldsymbol{\alpha}$, and $N$ source client $\mathcal{S}_1, \cdots, \mathcal{S}_N$, each with $K$ data batches $\{(\boldsymbol{X}_k^{\mathcal{S}_i}, \boldsymbol{Y}_k^{\mathcal{S}_i})\}_{k=1}^K$, define the *training error rate*

$$\hat{\varepsilon}(\boldsymbol{\alpha}) := \frac{1}{K} \sum_{i=1}^N \hat{\varepsilon}_i(\boldsymbol{\alpha})$$

Also, define the *expected testing error rate*

$$\varepsilon(\boldsymbol{\alpha}) := \mathbb{E}_{P^{\mathcal{S}_i} \sim \mathcal{Q}} \left[ \varepsilon_i(\boldsymbol{\alpha}) \mid P^{\mathcal{S}_i} \right] = \mathbb{E}_{P^{\mathcal{S}_i} \sim \mathcal{Q}} \mathbb{E}_{(\boldsymbol{X}_k^{\mathcal{S}_i}, \boldsymbol{Y}_k^{\mathcal{S}_i}) \sim P^{\mathcal{S}_i}} \left[ \hat{\varepsilon}_{ik}(\boldsymbol{\alpha}) \right]$$

*Remark* B.19. $\hat{\varepsilon}(\boldsymbol{\alpha})$ quantifies the averaged error rate across source clients' finite samples. $\varepsilon(\boldsymbol{\alpha})$ quantifies the expected error rate on a *new data batch* from a *new client* (target client). Noting that both error rates are defined with respect to the personalized model after adaptation.

### B.3.2 Generalization bound for one hypothesis

Next, we derive generalization bounds for one fixed adaptation rate $\boldsymbol{\alpha}$. Since we consider fixed $\boldsymbol{\alpha}$, for clarity, we denote

$$Z_{ik} := \hat{\varepsilon}_{ik}(\boldsymbol{\alpha})$$

$$\bar{Z}_{i\cdot} := \frac{1}{K}\sum_{k=1}^{K} Z_{ik} = \hat{\varepsilon}_i(\boldsymbol{\alpha})$$

$$\mu_i := \mathbb{E}_{(\boldsymbol{X}_k^{\mathcal{S}_i}, \boldsymbol{Y}_k^{\mathcal{S}_i}) \sim P^{\mathcal{S}_i}}[Z_{ik}] = \varepsilon_i(\boldsymbol{\alpha})$$

$$\bar{Z}_{\cdot\cdot} := \frac{1}{N}\sum_{i=1}^{N} \bar{Z}_{i\cdot} = \hat{\epsilon}(\boldsymbol{\alpha})$$

$$\bar{\mu}_\cdot := \frac{1}{N}\sum_{i=1}^{N} \mu_i$$

$$\mu := \mathbb{E}_{P^{\mathcal{S}_i} \sim \mathcal{Q}}\mu_i = \epsilon(\boldsymbol{\alpha})$$

Intuitively, with enough number of source clients and number of batches, we have $\bar{Z}_{\cdot\cdot} \approx \bar{\mu}_\cdot \approx \mu$.

**Lemma B.20** (Hoeffding's inequality). *Let $X_1, \cdots, X_n$ be independent random variables such that $a_i \leq X_i \leq b_i$ almost surely. Consider the sum of these random variables $S_n = X_1 + \cdots + X_n$. For all $\epsilon > 0$,*

$$\Pr(S_n - \mathbb{E}[S_n] \geq \epsilon) \leq \exp\left(-\frac{2\epsilon^2}{\sum_{i=1}^{n}(b_i - a_i)^2}\right)$$

*Proof.* Please refer to [13] $\qquad\square$

**Lemma B.21** (Concentration of averaged client expected error rates). *For any $\epsilon > 0$, we have*

$$\Pr(\bar{\mu}_\cdot - \mu \geq \epsilon) \leq \exp(-2N\epsilon^2)$$

*Proof.* Notice that $\mu_1, \cdots, \mu_N$ are independent given $\mathcal{Q}$. For all $i = 1, \cdots, N$, $\mathbb{E}\mu_i = \mu$ and $0 \leq \mu_i \leq 1$. Therefore,

$$\begin{aligned}
\Pr(\bar{\mu}_\cdot - \mu \geq \epsilon) &= \Pr\left(\sum_{i=1}^{N}\mu_i - N\mu \geq N\epsilon\right) \\
&= \Pr\left(\sum_{i=1}^{N}\mu_i - \mathbb{E}\left[\sum_{i=1}^{N}\mu_i\right] \geq N\epsilon\right) \\
&\leq \exp\left(-\frac{2\cdot(N\epsilon)^2}{N\cdot(1-0)^2}\right) \qquad \text{(Hoeffding's inequality)} \\
&= \exp(-2N\epsilon^2)
\end{aligned}$$

$\qquad\square$

**Lemma B.22** (Concentration of client empirical error rate). *For any $\epsilon > 0$,*

$$\Pr\left(\bar{Z}_{\cdot\cdot} - \bar{\mu}_\cdot \geq \epsilon\right) \leq \exp(-2NK\epsilon^2)$$

*Proof.* Given distributions $P^{\mathcal{S}_1}, \cdots, P^{\mathcal{S}_N}$, we have $Z_{11}, \cdots, Z_{1K}, Z_{21}, \cdots, Z_{NK}$ are independent. For any $i = 1, \cdots, N$ and $k = 1, \cdots, K$, we have $\mathbb{E}_{(\boldsymbol{X}_k^{\mathcal{S}_i}, \boldsymbol{Y}_k^{\mathcal{S}_i}) \sim P^{\mathcal{S}_i}} Z_{ik} = \mu_i$ and $0 \leq Z_{ik} \leq 1$. Therefore,

$$\Pr\left(\bar{Z}_{\cdot\cdot} - \bar{\mu}_\cdot \geq \epsilon \mid P^{\mathcal{S}_1}, \cdots, P^{\mathcal{S}_N}\right) = \Pr\left(\sum_{i=1}^{N}\sum_{k=1}^{K} Z_{ik} - \sum_{i=1}^{N} K\mu_i \geq NK\epsilon \,\middle|\, P^{\mathcal{S}_1}, \cdots, P^{\mathcal{S}_N}\right)$$

$$= \Pr\left(\sum_{i=1}^{N}\sum_{k=1}^{K} Z_{ik} - \mathbb{E}\left[\sum_{i=1}^{N}\sum_{k=1}^{K} Z_{ik}\right] \geq NK\epsilon \,\middle|\, P^{\mathcal{S}_1}, \cdots, P^{\mathcal{S}_N}\right)$$

$$\leq \exp\left(-\frac{2 \cdot (NK\epsilon)^2}{NK \cdot (1-0)^2}\right) \qquad \text{(Hoeffding's inequality)}$$

$$= \exp(-2NK\epsilon^2)$$

Then, we use the tower property,

$$\Pr(\bar{Z}_{..} - \bar{\mu}_. \geq \epsilon) = \mathbb{E}_{P^{\mathcal{S}_1}, \cdots, P^{\mathcal{S}_N} \overset{\text{i.i.d.}}{\sim} \mathcal{Q}} \Pr(\bar{Z}_{..} - \bar{\mu}_. \geq \epsilon \mid P^{\mathcal{S}_1}, \cdots, P^{\mathcal{S}_N})$$

$$\leq \sup_{P^{\mathcal{S}_1}, \cdots, P^{\mathcal{S}_N}} \Pr(\bar{Z}_{..} - \bar{\mu}_. \geq \epsilon \mid P^{\mathcal{S}_1}, \cdots, P^{\mathcal{S}_N})$$

$$\leq \exp(-2NK\epsilon^2)$$

$\square$

**Proposition B.23** (Generalization for one hypothesis)**.** *For any fixed global model $\mathbf{w}_G$ and adaptation rate $\boldsymbol{\alpha}$, for any $\epsilon > 0$, we have*

$$\Pr\left(|\hat{\varepsilon}(\boldsymbol{\alpha}) - \varepsilon(\boldsymbol{\alpha})| \geq \epsilon\right) \leq 4\exp\left(-\frac{2NK\epsilon^2}{(\sqrt{K}+1)^2}\right)$$

*Proof.* For any $\epsilon > 0$, we have

$$\Pr(\bar{Z}_{..} - \mu \geq \epsilon) = \Pr((\bar{Z}_{..} - \bar{\mu}_.) + (\bar{\mu}_. - \mu) \geq \epsilon)$$

$$\leq \inf_{\epsilon'}\left[\Pr((\bar{Z}_{..} - \bar{\mu}_. \geq \epsilon') \vee (\bar{\mu}_. - \mu \geq \epsilon - \epsilon'))\right]$$

$$\leq \inf_{\epsilon'}\left[\Pr(\bar{Z}_{..} - \bar{\mu}_. \geq \epsilon') + \Pr(\bar{\mu}_. - \mu \geq \epsilon - \epsilon')\right]$$

$$\leq \inf_{\epsilon'}[\exp(-2NK(\epsilon')^2) + \exp(-2N(\epsilon - \epsilon')^2)] \qquad \text{(Lemma B.21 and B.22)}$$

To make the bound clear (although not optimal), we choose $\epsilon' = \frac{1}{\sqrt{K}+1}\epsilon$ and thus $\epsilon - \epsilon' = \frac{\sqrt{K}}{\sqrt{K}+1}\epsilon$. Then the bound becomes,

$$\Pr(\bar{Z}_{..} - \mu \geq \epsilon) \leq 2\exp\left(-\frac{2NK\epsilon^2}{(\sqrt{K}+1)^2}\right)$$

Similarly we can show that

$$\Pr(\bar{Z}_{..} - \mu \leq -\epsilon) \leq 2\exp\left(-\frac{2NK\epsilon^2}{(\sqrt{K}+1)^2}\right)$$

Therefore,

$$\Pr\left(|\hat{\varepsilon}(\boldsymbol{\alpha}) - \varepsilon(\boldsymbol{\alpha})| \geq \epsilon\right) = \Pr\left(|\bar{Z}_{..} - \mu| \geq \epsilon\right) \leq 4\exp\left(-\frac{2NK\epsilon^2}{(\sqrt{K}+1)^2}\right)$$

$\square$

*Remark* B.24. The RHS is function of both (1) $N$, the number of source clients and (2) $K$, the number of data batches on each client.

- When $N \to \infty$, given any fixed $K \geq 1$, the RHS $\to 0$, indicating that $\hat{\varepsilon}(\boldsymbol{\alpha}) \overset{p}{\to} \varepsilon(\boldsymbol{\alpha})$ (convergence in probability).

- However, given a fixed finite $N$, when $K \to \infty$, the RHS does not limit to zero. Intuitively, sampling more batches on finite source clients only help the algorithm learn finite distribution $P^{\mathcal{S}_1}, \cdots, P^{\mathcal{S}_N}$. However, more data batches on existing clients does not help further exploration of the meta-distribution $\mathcal{Q}$ and generalization to novel target clients. Actually, given a fixed finite $N$, when $K \to \infty$, $\hat{\varepsilon}(\boldsymbol{\alpha}) \overset{p}{\to} \frac{1}{N}\sum_{i=1}^{N} \varepsilon_i(\boldsymbol{\alpha}) \neq \varepsilon(\boldsymbol{\alpha})$.

- If we put data from $N$ sources (each with $K$ batches) together as one source with $NK$ batches. The generalization bound is looser.

### B.3.3 Generalization bound for hypothesis space (proof of Theorem 5.1)

Finally, we derive the generalization bound for the hypothesis space. We first show in Lemma B.27 that $\hat{\varepsilon}_{ij}(\boldsymbol{\alpha})$ is $(LH)$-Lipschitz to $\boldsymbol{\alpha}$, then we apply standard generalization analysis in Theorem 5.1 based on covering number [32].

**Definition B.25** ($L$-Lipschitz)**.** The neural network $f(\boldsymbol{x}; \boldsymbol{w})$ is $L$-Lipschitz w.r.t. $\boldsymbol{w}$, if $\forall \boldsymbol{x}$ and $\boldsymbol{w}_1, \boldsymbol{w}_2$.

$$\|f(\boldsymbol{x}; \boldsymbol{w}_1) - f(\boldsymbol{x}; \boldsymbol{w}_2)\|_2 \leq L \cdot \|\boldsymbol{w}_1 - \boldsymbol{w}_2\|_2$$

**Definition B.26** ($H$-module-wise-bounded update direction)**.** The update direction is $H$-module-wise-bounded for a data batch $\boldsymbol{X}_k^{\mathcal{S}_i}$ if

$$\|(\boldsymbol{h}_k^{\mathcal{S}_i})^{[l]}\|_2 \leq H, \quad \forall l = 1, \cdots, d$$

where $(\boldsymbol{h}_k^{\mathcal{S}_i})^{[l]}$ is the update direction corresonding to the $l$-th module and $d$ is the number of modules in the neural network.

**Lemma B.27** (Lipschitz error rate)**.** *Given a data batch $(\boldsymbol{X}_k^{\mathcal{S}_i}, \boldsymbol{Y}_k^{\mathcal{S}_i})$, if the update direction is $H$-module-wise-bounded, given any two adaptation rates $\boldsymbol{\alpha}_1, \boldsymbol{\alpha}_2$ and the global model $\boldsymbol{w}_G$, we have*

$$|\hat{\varepsilon}_{ij}(\boldsymbol{\alpha}_1) - \hat{\varepsilon}_{ij}(\boldsymbol{\alpha}_2)| \leq LH \cdot \|\boldsymbol{\alpha}_1 - \boldsymbol{\alpha}_2\|_2$$

*Proof.*

$$
\begin{aligned}
&|\hat{\varepsilon}_{ij}(\boldsymbol{\alpha}_1) - \hat{\varepsilon}_{ij}(\boldsymbol{\alpha}_2)| \\
&= \left| \left( \frac{1}{B} \sum_{b=1}^{B} \left( 1 - (\boldsymbol{y}_{k,b}^{\mathcal{S}_i})^{\top} f(\boldsymbol{x}_{k,b}^{\mathcal{S}_i}; \boldsymbol{w}_k^{\mathcal{S}_i}(\boldsymbol{\alpha}_1)) \right) \right) - \left( \frac{1}{B} \sum_{b=1}^{B} \left( 1 - (\boldsymbol{y}_{k,b}^{\mathcal{S}_i})^{\top} f(\boldsymbol{x}_{k,b}^{\mathcal{S}_i}; \boldsymbol{w}_k^{\mathcal{S}_i}(\boldsymbol{\alpha}_2)) \right) \right) \right| \\
&= \left| \frac{1}{B} \sum_{b=1}^{B} (\boldsymbol{y}_{k,b}^{\mathcal{S}_i})^{\top} \left( f(\boldsymbol{x}_{k,b}^{\mathcal{S}_i}; \boldsymbol{w}_k^{\mathcal{S}_i}(\boldsymbol{\alpha}_1)) - f(\boldsymbol{x}_{k,b}^{\mathcal{S}_i}; \boldsymbol{w}_k^{\mathcal{S}_i}(\boldsymbol{\alpha}_2)) \right) \right| \\
&\leq \frac{1}{B} \sum_{b=1}^{B} \|\boldsymbol{y}_{k,b}^{\mathcal{S}_i}\|_2 \cdot \left\| f(\boldsymbol{x}_{k,b}^{\mathcal{S}_i}; \boldsymbol{w}_k^{\mathcal{S}_i}(\boldsymbol{\alpha}_1)) - f(\boldsymbol{x}_{k,b}^{\mathcal{S}_i}; \boldsymbol{w}_k^{\mathcal{S}_i}(\boldsymbol{\alpha}_2)) \right\|_2 \\
&\leq \frac{1}{B} \sum_{b=1}^{B} \|\boldsymbol{y}_{k,b}^{\mathcal{S}_i}\|_2 \cdot L \cdot \left\| \boldsymbol{w}_k^{\mathcal{S}_i}(\boldsymbol{\alpha}_1) - \boldsymbol{w}_k^{\mathcal{S}_i}(\boldsymbol{\alpha}_2) \right\|_2 && (L\text{-Lipschitz model}) \\
&\leq \frac{1}{B} \sum_{b=1}^{B} \|\boldsymbol{y}_{k,b}^{\mathcal{S}_i}\|_2 \cdot L \cdot H \cdot \|\boldsymbol{\alpha}_1 - \boldsymbol{\alpha}_2\|_2 && (\text{Lemma B.8}) \\
&= LH \cdot \|\boldsymbol{\alpha}_1 - \boldsymbol{\alpha}_2\|_2
\end{aligned}
$$

$\square$

*Remark* B.28. Intuitively, Lemma B.27 shows that small change in $\boldsymbol{\alpha}$ will result in bounded change on $\hat{\varepsilon}_{ij}(\boldsymbol{\alpha})$.

**Corollary B.29.** $\hat{\epsilon}(\boldsymbol{\alpha})$ *and* $\epsilon(\boldsymbol{\alpha})$ *are* $(LH)$-*Lipschitz w.r.t.* $\boldsymbol{\alpha}$.

*Proof.* $\hat{\epsilon}(\boldsymbol{\alpha})$ and $\epsilon(\boldsymbol{\alpha})$ are expectations of $\hat{\varepsilon}_{ij}(\boldsymbol{\alpha})$ given the empirical and expected distribution of $(\boldsymbol{X}_k^{\mathcal{S}_i}, \boldsymbol{Y}_k^{\mathcal{S}_i})$. Lipschitz property is preserved. $\square$

**Theorem 5.1** (Generalization for hypothesis space)**.** Let $\mathcal{H} = \{\boldsymbol{\alpha} : \|\boldsymbol{\alpha}\|_2 \leq R\}$ be the hypothesis space (space of adaptation rates), $N$ be the number of source clients, and $K$ be the number of data batches on each source client. Assuming (1) $L$-Lipschitz model, and (2) $H$-module-wise-bounded update direction. For any fixed global model $\boldsymbol{w}_G$ and any $\epsilon > 0$, we have

$$\Pr(\sup_{\boldsymbol{\alpha} \in \mathcal{H}} |\varepsilon(\boldsymbol{\alpha}) - \hat{\varepsilon}(\boldsymbol{\alpha})| \geq \epsilon) \leq \left( \frac{12LHR}{\epsilon} \right)^d \cdot 4 \exp \left( -\frac{NK\epsilon^2}{2(\sqrt{K}+1)^2} \right) \tag{10}$$

where $\hat{\varepsilon}(\boldsymbol{\alpha})$ is the average **post-adaptation** error rate on source clients, and $\varepsilon(\boldsymbol{\alpha})$ is the expected **post-adaptation** error rate on clients' population.

*Proof.* We use covering number to derive the generalization bound [32]. Define estimation error

$$\Delta_\epsilon(\boldsymbol{\alpha}) = \varepsilon(\boldsymbol{\alpha}) - \hat\varepsilon(\boldsymbol{\alpha})$$

Then,

$$
\begin{aligned}
|\Delta_\epsilon(\boldsymbol{\alpha}_1) - \Delta_\epsilon(\boldsymbol{\alpha}_2)| &= |[\varepsilon(\boldsymbol{\alpha}_1) - \hat\varepsilon(\boldsymbol{\alpha}_1)] - [\varepsilon(\boldsymbol{\alpha}_2) - \hat\varepsilon(\boldsymbol{\alpha}_2)]| \\
&\leq |\varepsilon(\boldsymbol{\alpha}_1) - \varepsilon(\boldsymbol{\alpha}_2)| + |\hat\varepsilon(\boldsymbol{\alpha}_1) - \hat\varepsilon(\boldsymbol{\alpha}_2)| \\
&\leq 2LH \cdot \|\boldsymbol{\alpha}_1 - \boldsymbol{\alpha}_2\|_2 \qquad\qquad \text{(Corollary B.29)}
\end{aligned}
$$

$\mathcal{A} = \{\boldsymbol{\alpha} : \|\boldsymbol{\alpha}\|_2 \leq R, \boldsymbol{\alpha} \in \mathbb{R}^d\}$ can be covered by $K = \mathcal{N}_2(R, r)$ L2 balls with radius $r = \frac{\epsilon}{4LH}$. Lemma 6.27 in [32] shows that

$$S = \mathcal{N}_2(R, r) \leq \left(\frac{3R}{r}\right)^d = \left(\frac{12LHR}{\epsilon}\right)^d$$

Denote these L2 balls to be $B_1, \cdots, B_S$,

$$\Pr\left(\sup_{\boldsymbol{\alpha} \in \mathcal{A}} |\varepsilon(\boldsymbol{\alpha}) - \hat\varepsilon(\boldsymbol{\alpha})| \geq \epsilon\right) \leq \sum_{s=1}^{S} \Pr\left(\sup_{\boldsymbol{\alpha} \in B_s} |\varepsilon(\boldsymbol{\alpha}) - \hat\varepsilon(\boldsymbol{\alpha})| \geq \epsilon\right)$$

For each ball $B_s$, $s = 1, \cdots, S$, denote the center to be $\boldsymbol{\alpha}_s$. For any $\boldsymbol{\alpha} \in B_s$, we have $\|\boldsymbol{\alpha} - \boldsymbol{\alpha}_s\| \leq \frac{\epsilon}{4LH}$, therefore

$$|\Delta_\epsilon(\boldsymbol{\alpha}_s) - \Delta_\epsilon(\boldsymbol{\alpha})| \leq 2LH \cdot \|\boldsymbol{\alpha} - \boldsymbol{\alpha}_s\| \leq \frac{\epsilon}{2}$$

Intuitively, every $\boldsymbol{\alpha} \in B_s$ has similar error rate. Therefore, the error rate for the whole ball is upper bounded, as long as the center $\boldsymbol{\alpha}_s$ has a small error rate

$$
\begin{aligned}
\Pr\left(\sup_{\boldsymbol{\alpha} \in B_s} |\varepsilon(\boldsymbol{\alpha}) - \hat\varepsilon(\boldsymbol{\alpha})| \geq \epsilon\right) &= \Pr\left(\sup_{\boldsymbol{\alpha} \in B_s} |\Delta_\epsilon(\boldsymbol{\alpha})| \geq \epsilon\right) \\
&\leq \Pr\left(\sup_{\boldsymbol{\alpha} \in B_s} [|\Delta_\epsilon(\boldsymbol{\alpha}_s)| + |\Delta_\epsilon(\boldsymbol{\alpha}_s) - \Delta_\epsilon(\boldsymbol{\alpha})|] \geq \epsilon\right) \\
&\leq \Pr\left(\sup_{\boldsymbol{\alpha} \in B_s} |\Delta_\epsilon(\boldsymbol{\alpha}_s)| + \frac{\epsilon}{2} \geq \epsilon\right) \\
&= \Pr\left(|\varepsilon(\boldsymbol{\alpha}_s) - \hat\varepsilon(\boldsymbol{\alpha}_s)| \geq \frac{\epsilon}{2}\right)
\end{aligned}
$$

Finally, by Proposition B.23, for each $\boldsymbol{\alpha}_s$

$$\Pr\left(|\varepsilon(\boldsymbol{\alpha}_s) - \hat\varepsilon(\boldsymbol{\alpha}_s)| \geq \frac{\epsilon}{2}\right) \leq 4\exp\left(-\frac{NK\epsilon^2}{2(\sqrt{K}+1)^2}\right)$$

Put all together

$$\Pr\left(\sup_{\boldsymbol{\alpha} \in \mathcal{A}} |\varepsilon(\boldsymbol{\alpha}) - \hat\varepsilon(\boldsymbol{\alpha})| \geq \epsilon\right) \leq \left(\frac{12LHR}{\epsilon}\right)^d \cdot 4\exp\left(-\frac{NK\epsilon^2}{2(\sqrt{K}+1)^2}\right)$$

$\square$

## C Additional experiments

### C.1 Detailed experiment settings

#### C.1.1 CIFAR-10 experiments

**Data preparation** We use a benchmarking three-way split [50]: we randomly split the dataset to 300 clients, 240 of them are source clients and 60 are target clients. Each source client has 160 training samples and 40 validation samples, while each target client has 200 testing samples. We simulate three kinds of distribution shifts: feature shift, label shift, and hybrid shift. For feature shift, we follow [12, 17], randomly apply 15 different kinds of corruptions to the source clients (Figure 7a), and 4 new kinds of corruptions to the target clients (Figure 7b) to test the generalization of ATP. The corruption severity is randomly selected from $\{1, 2, 3, 4, 5\}$. For label shift, we use the step partition [5], where each client has 8 minor classes with 5 images per class, and 2 major classes with 80 images per class. For the hybrid shift, we apply both step partition and feature corruptions.

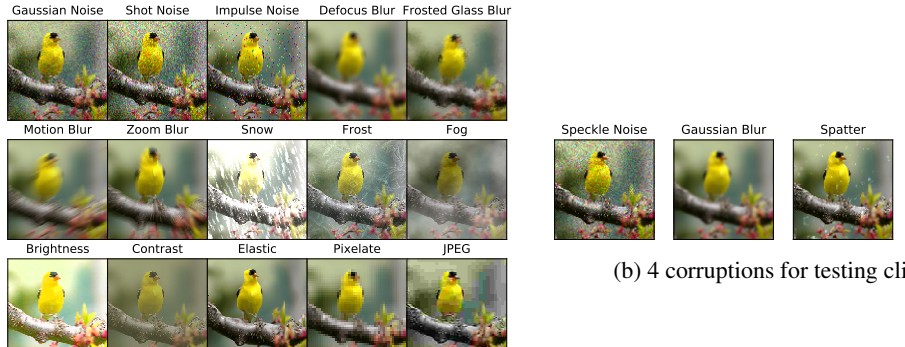

(a) 15 corruptions for training clients

(b) 4 corruptions for testing clients

Figure 7: $15 + 4$ different corruptions we use to construct feature shift

**Global model training** We first train a global model with FedAvg [31] over the training sets of source clients.

- ResNet-18: The global model is ResNet-18 with ImageNet pretrained parameter (provided by `torchvision`). We train the global model for $T = 200$ communication rounds with full participation (cohort size $C = 240$), local epochs $E = 1$, learning rate $\eta = 0.01$ and batch size $B = 20$.
- Shallow CNN: The global model is a randomly initialized 5-layer CNN. We train the global model for $T = 200$ communication rounds with full participation (cohort size $C = 240$), local epochs $E = 1$, learning rate $\eta = 0.1$ and batch size $B = 20$.

`ATP` **training** We initialize the adaptation rates as a all-zero vector, and optimize it over the validation sets of source clients. We optimize the adaptation rates for $T = 200$ (for ResNet-18) or $400$ (for Shallow CNN) communication rounds with partial participation (cohort size $C = 60$), learning rate $\eta = 0.1$ and batch size $B = 20$.

`ATP` **testing** We test the optimized adaptation rates on each target client. We use batch size $B = 20$ by default, and test different batch size in Subsection 6.3.

#### C.1.2 CIFAR-100 experiments

**Data preparation** The data preparation is similar to CIFAR-10 experiments. The only difference is for label shift, each client has 98 minor classes with 1 image per class, and 2 major classes with 51 images per class. Same partition is applied to hybrid shift.

**Global model training** We first train a global model with FedAvg [31] over the training sets of source clients. The global model is ResNet-18 with ImageNet pretrained parameter (provided by `torchvision`). We train the global model for $T = 200$ communication rounds with full participation (cohort size $C = 240$), local epochs $E = 1$, learning rate $\eta = 0.01$ and batch size $B = 20$.

`ATP` **training** We initialize the adaptation rates as a all-zero vector, and optimize it over the validation sets of source clients. We optimize the adaptation rates for $T = 200$ communication rounds with partial participation (cohort size $C = 60$), learning rate $\eta = 0.1$ and batch size $B = 20$.

`ATP` **testing** We test the optimized adaptation rates on each target client. We use batch size $B = 20$.

### C.1.3 Digits-5 experiments

**Data preparation** Digits-5 dataset contains five domains: MNIST, SVHN, USPS, SynthDigits, and MNIST-M. We adopt the leave-one-domain-out evaluation protocol [10], i.e., one domain is chosen as the held-out testing domain, and the remaining domains are regarded as source training domains. We follow the data preprocessing in [25], while additionally applying step partition to inject label shift. Each domain is divided into 10 clients, leading to a total of 40 source clients and 10 target clients. Consequently, each client ends up with approximately 743 images spread across 10 classes. Each source client has $80\%$ of its samples as training set and the remained $20\%$ as testing set. Each client has 2 major classes and 8 minor class, where the ratio of images per class is approximately $16 : 1$ (the same as our CIFAR-10 experiments). Since there is already domain shift, we do not add corruptions.

**Global model training** We first train a global model with FedAvg [31] over the training sets of source clients. The global model is ResNet-18 with ImageNet pretrained parameter (provided by `torchvision`). We train the global model for $T = 200$ communication rounds with full participation (cohort size $C = 50$), local epochs $E = 1$, learning rate $\eta = 0.01$ and batch size $B = 20$.

`ATP` **training** We initialize the adaptation rates as a all-zero vector, and optimize it over the validation sets of source clients. We optimize the adaptation rates for $T = 200$ communication rounds with partial participation (cohort size $C = 10$), learning rate $\eta = 0.5$ and batch size $B = 200$.

`ATP` **testing** We test the optimized adaptation rates on each target client. We use batch size $B = 200$.

### C.1.4 PACS experiments

**Data preparation** PACS dataset contains four domains: art, cartoon, photo, and sketch. We adopt the leave-one-domain-out evaluation protocol [10], i.e., one domain is chosen as the held-out testing domain, and the remaining domains are regarded as source training domains. We follow the data preprocessing in [10], while additionally applying step partition to inject label shift. Each domain is divided into 7 clients, leading to a total of 21 source clients and 7 target clients. Each source client has $80\%$ of its samples as training set and the remained $20\%$ as testing set. Each client has 2 major classes and 5 minor class, where the ratio of images per class is approximately $16 : 1$ (the same as our CIFAR-10 experiments). Since there is already domain shift, we do not add corruptions.

**Global model training** We first train a global model with FedAvg [31] over the training sets of source clients. The global model is ResNet-18 with ImageNet pretrained parameter (provided by `torchvision`). We train the global model for $T = 200$ communication rounds with full participation (cohort size $C = 21$), local epochs $E = 1$, learning rate $\eta = 0.05$ and batch size $B = 20$.

`ATP` **training** We initialize the adaptation rates as a all-zero vector, and optimize it over the validation sets of source clients. We optimize the adaptation rates for $T = 500$ communication rounds with full participation (cohort size $C = 21$), learning rate $\eta = 0.5$ and batch size $B = 200$.

`ATP` **testing** We test the optimized adaptation rates on each target client. We use batch size $B = 200$.

### C.1.5 Algorithm details

**Assignment matrix $A$** In the main test, we mentioned that $A \in \mathbb{R}^{D \times d}$ is a $0 - 1$ assignment matrix that maps each adaptation rate $\alpha^{[l]}$ to the indices of the $l$-th module's parameters in $w$. Mathematically,

$$A_{kl} = \begin{cases} 1, & \text{if the } k\text{-th parameter in } w \text{ belongs to the } l\text{-th module} \\ 0, & \text{otherwise} \end{cases}$$

If there are $d = 3$ modules, each with 1, 2, and 3 parameters, so $D = 1+2+3 = 6$, the corresponding assignment matrix will be

$$A = \begin{bmatrix} 1 & 0 & 0 \\ 0 & 1 & 0 \\ 0 & 1 & 0 \\ 0 & 0 & 1 \\ 0 & 0 & 1 \\ 0 & 0 & 1 \end{bmatrix}$$

**Computation**

We did our experiments with single NVIDIA Tesla V100 GPU. However, our experiment should only require less than 2GB of GPU memory.

## C.2 Compatibility to model architecture (RQ1)

In this part, we evaluate `ATP` with two more model architectures: a 5-layer Shallow CNN as a smaller model and ResNet-50 as a larger model.

Table 5: `ATP` with different model architectures, accuracy (mean ± s.d. %) on target clients

| Method | Shallow CNN on CIFAR-10 | | | | ResNet-50 on CIFAR-100 | | | |
|---|---|---|---|---|---|---|---|---|
| | Feature shift | Label shift | Hybrid shift | Avg. Rank | Feature shift | Label shift | Hybrid shift | Avg. Rank |
| No adaptation | 64.39 ± 0.18 | 69.33 ± 0.37 | 61.99 ± 0.47 | 7.3 | 45.31 ± 0.30 | 51.63 ± 0.15 | 40.01 ± 0.17 | 7.3 |
| BN-Adapt | 66.46 ± 0.22 | 54.99 ± 0.38 | 50.40 ± 0.43 | 7.0 | 47.75 ± 0.29 | 34.85 ± 0.26 | 30.31 ± 0.09 | 7.3 |
| SHOT | 65.60 ± 0.18 | 49.98 ± 0.29 | 45.95 ± 0.47 | 9.0 | 45.42 ± 0.30 | 31.06 ± 0.32 | 27.44 ± 0.14 | 9.3 |
| Tent | 65.61 ± 0.24 | 50.12 ± 0.25 | 45.91 ± 0.49 | 8.7 | 45.91 ± 0.46 | 31.34 ± 0.11 | 27.93 ± 0.31 | 8.3 |
| T3A | 64.31 ± 0.27 | 66.96 ± 0.43 | 59.65 ± 0.58 | 8.3 | 45.31 ± 0.30 | 51.42 ± 0.15 | 39.89 ± 0.20 | 7.7 |
| MEMO | 65.89 ± 0.31 | 71.95 ± 0.25 | 64.17 ± 0.47 | 5.3 | 48.42 ± 0.14 | 55.19 ± 0.28 | 42.53 ± 0.20 | 3.7 |
| EM | 61.74 ± 0.25 | 76.28 ± 0.29 | 67.54 ± 0.41 | 5.0 | 43.00 ± 0.31 | 59.34 ± 0.15 | 44.82 ± 0.27 | 5.0 |
| BBSE | 56.92 ± 0.53 | 75.99 ± 0.44 | 66.64 ± 0.53 | 6.3 | 37.26 ± 0.64 | 56.97 ± 0.20 | 40.09 ± 0.51 | 7.0 |
| Surgical | 64.45 ± 0.12 | 73.75 ± 0.42 | 65.67 ± 0.44 | 5.7 | 45.18 ± 0.38 | 54.83 ± 0.26 | 42.50 ± 0.33 | 6.7 |
| `ATP`-batch | 66.90 ± 0.05 | 76.23 ± 0.32 | 68.88 ± 0.35 | 2.3 | 48.35 ± 0.45 | 58.06 ± 0.53 | 46.82 ± 0.32 | 2.7 |
| `ATP`-online | **67.13 ± 0.17** | **78.56 ± 0.32** | **71.52 ± 0.51** | **1.0** | **49.08 ± 0.26** | **61.86 ± 0.25** | **49.51 ± 0.23** | **1.0** |

From Table 5, we observe that under the new model architecture (and the new dataset), the performance of `ATP` is highly similar to the results of the ResNet-18 + CIFAR10 experiment in Table 1 in Subsection 6.1. `ATP`, in all three scenarios, can handle various types of distribution shifts and surpass baseline methods. This suggests that `ATP` is compatible with multiple model architectures.

## C.3 Robustness to global model

In this subsection, we design experiments to answer the following question: *is ATP robust to the choice of global model?* Specifically, we have three sub-questions:

- Is ATP robust to the parameter of global model? (C.3.1)
- Is ATP robust to the algorithm to train global model? (C.3.2)

### C.3.1 Robustness to the parameter of global model (online updated global model)

In the main text, we primarily focused on the scenario where the global model remains fixed. However, in practical FL systems, the global model may also undergo continuous online updates. Therefore, after obtaining the adaptation rates through ATP training, the global model might have been further updated for several rounds. This raises a question: *Are the "outdated" adaptation rates still effective after several rounds of updates to the global model?*

Table 6: Accuracy (%), ATP can learn adaptation rates that generalize to global models with different numbers of communication rounds under hybrid shift on CIFAR-10

| Method | $200 + 0$ Rounds | $+10$ Rounds | $+20$ Rounds | $+50$ Rounds | $+100$ Rounds |
|---|---|---|---|---|---|
| No adaptation | $63.68 \pm 0.24$ | $63.88 \pm 0.20$ | $64.03 \pm 0.13$ | $64.30 \pm 0.08$ | $64.56 \pm 0.11$ |
| ATP-batch | $73.05 \pm 0.35$ | $73.20 \pm 0.40$ | $73.25 \pm 0.37$ | $73.47 \pm 0.48$ | $73.61 \pm 0.28$ |
| ATP-online | $75.37 \pm 0.22$ | $75.61 \pm 0.23$ | $75.69 \pm 0.20$ | $75.80 \pm 0.15$ | $75.83 \pm 0.28$ |

We design experiment to apply the "outdated" adaptation rates to the global model that has undergone additional updates for several rounds, to see if they can still improve the test-time accuracy of the global model. Specifically, we optimize the adaptation rates $\alpha$ with $w_G^T$ where $T = 200$, but test the adaptation rates with $w_G^{T+\Delta T}$ with $\Delta T = 10, 20, 50, 100$ rounds. We use the same setting of hybrid shift on CIFAR-10 experiments. As shown in Table 6, while further optimizing the global model can marginally improve the accuracy, both ATP-batch and ATP-online can effectively enhance the test-time accuracy through personalization, even when $\alpha$ is trained using an outdated version of the global model.

### C.3.2 Robustness to the algorithm to train global model

In the main text, we used FedAvg [31] to train the global model. However, in real-world FL systems, other FL algorithms may be employed for training the global model, considering stability optimization or fairness. Therefore, we aim to investigate *whether ATP can also be applied to other commonly used FL algorithms*.

Table 7: Accuracy (%), ATP enhances different global models under hybrid shift on CIFAR-10

| Method | FedAvg | FedProx ($\mu = 0.01$) | $q$-FFL ($q = 1$) |
|---|---|---|---|
| No adaptation | $63.68 \pm 0.24$ | $63.77 \pm 0.25$ | $63.87 \pm 0.23$ |
| ATP-batch | $73.05 \pm 0.35$ | $72.95 \pm 0.33$ | $73.15 \pm 0.21$ |
| ATP-online | $75.37 \pm 0.22$ | $75.51 \pm 0.19$ | $75.79 \pm 0.15$ |

In particular, we use FedProx [23], an FL algorithm designed to handle heterogeneous setting, and $q$-FFL [24], an FL algorithm enhancing performance fairness among participating clients. For all global model, we use the same setting of hybrid shift on CIFAR-10 experiments. As shown in Table 7, both ATP-batch and ATP-online can consistently improve the test-time accuracy across different FL algorithms to train global models.

## C.4 Convergence and generalization

In Section 5, Appendix B.2 and B.3, we theoretically show that `ATP` has good convergence and generalization guarantees. In this section, we visualize the training and testing loss curves to verify the fast convergence and superior generalization of `ATP` under different cohort size $C$. The results are shown in Figure 8.

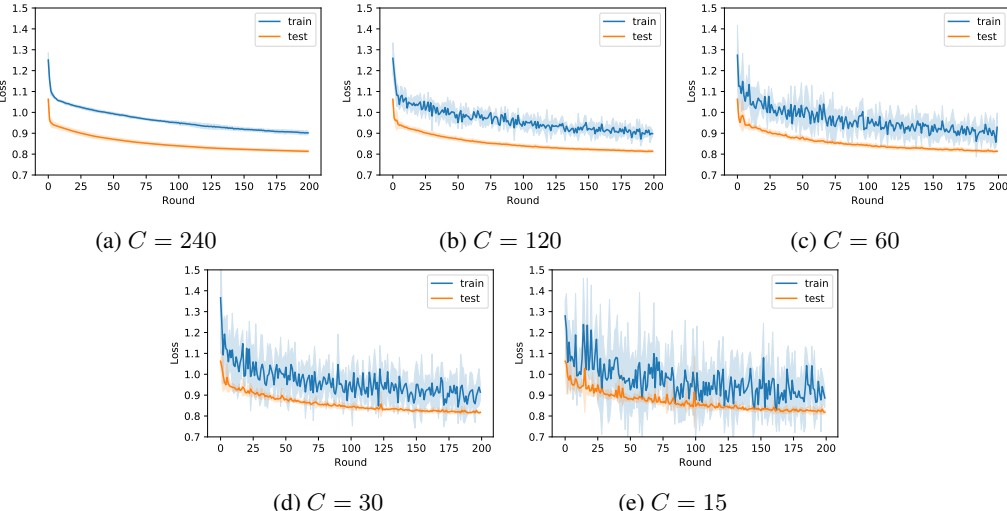

(a) $C = 240$ (b) $C = 120$ (c) $C = 60$

(d) $C = 30$ (e) $C = 15$

Figure 8: Loss curves of `ATP` under different cohort size $C$

**Convergence** Under full participation ($C = 240$), both the training and testing loss converge stably and fast, indicating the reliable convergence of `ATP`. With partial participation, as the cohort size decreases ($C = 120, 60, 30, 15$), the training loss curve exhibits greater fluctuations, primarily due to sampling different subsets of clients in each communication round. However, the testing loss curve still converge stably with similar speed, indicating that `ATP` is robust to partial participation.

**Generalization** Under full participation ($C = 240$), the training and testing loss curves decrease synchronously without any overfitting. This implies that our algorithm exhibits excellent generalization. Similar observations can be made for partial participation ($C = 120, 60, 30, 15$). Additionally, it is worth noting that the test loss is lower than the train loss, which may seem counterintuitive. This is primarily due to the use of different corruptions between the testing and source clients. The accuracy of clients varies significantly under different corruptions, as evidenced by the fluctuations in the training curve when $C = 15$. However, we can still analyze the generalization performance by comparing the trends of the two curves.

## C.5 Toy example for negative adaptation rate (RQ2)

In Section 6.2, we notice that `ATP` learns negative adaptation rates for running means and variance under label shift. In this subsection, we use a toy example to show why negative adaptation rate can improve the model performance under label distribution shift.

We consider a binary classification problem with input $x \in \mathbb{R}$ and binary output $y \in \{-1, +1\}$, where $-1$ is the negative class and $+1$ is the positive class. Let the feature for negative samples $(x|y = -1) \sim \mathcal{N}(-1, 0.8^2)$ and for positive samples $(x|y = +1) \sim \mathcal{N}(+1, 0.8^2)$. Let the label distribution $\Pr(y = 1) = \frac{1}{2}$ for training set, and $\Pr(y = 1) = \frac{5}{6}$ for testing set. Therefore, for the training distribution, we have

$$\mathbb{E}x = \Pr(y = -1)\mathbb{E}(x|y = -1) + \Pr(y = +1)\mathbb{E}(x|y = +1) = 0$$
$$Var(x) = \mathbb{E}[Var(x|y)] + Var(\mathbb{E}[x|y]) = 1.64$$

We consider a simple network with only one BN layer, with both normalization and affine transformation (as a linear classifier). There are four modules, each is a scalar: running mean $\mu$, running variance $\sigma^2$, weight $\gamma$, bias $\beta$.

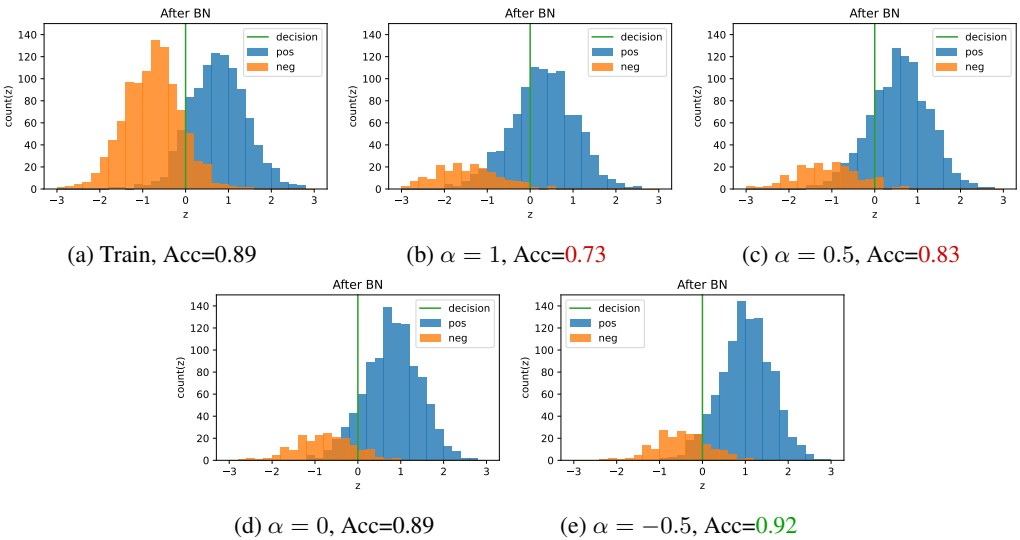

| (a) Train, Acc=0.89 | (b) $\alpha = 1$, Acc=0.73 | (c) $\alpha = 0.5$, Acc=0.83 |

| (d) $\alpha = 0$, Acc=0.89 | (e) $\alpha = -0.5$, Acc=0.92 |

Figure 9: Adapting batch norm running statistics under label shift.

**Training**  During training, given enough training data, we have $\mu_{train} = \mathbb{E}x = 0$ and $\sigma^2_{train} = 1.64$. Figure 9a shows the histogram of $z = \frac{x - \mu}{\sigma}$, i.e., the intermediate feature after normalization before the transformation. By comparing the histograms of $z$ of two classes, we notice that the optimal decision boundary is $z = 0$, which indicate that $\beta_{train} = 0$ and $\gamma_{train} > 0$. We store the corresponding $\mu_{train}, \sigma^2_{train}, \gamma_{train}, \beta_{train}$, and only update running statistics $\mu_{train}, \sigma^2_{train}$ during testing.

**Testing without updating running statistics ($\alpha = 0$)**  Figure 9d shows the testing result when we do not update the running statistics, i.e., $\alpha = 0$. Since two conditional feature distributions are symmetric, the accuracy will not change.

**Testing with $\alpha > 0$**  Positive adaptation rates align the intermediate feature distribution. When we use $\alpha = 1$, the distribution of $z$ will be centralized. As shown in Figure 9b, such alignment greatly reduces the accuracy. Similar result is also observed with any positive $\alpha$, e.g., $\alpha = 0.5$ in Figure 9c.

**Testing with $\alpha < 0$**  While $\alpha = 0$ has stable accuracy under label shift, by comparing the histograms of $z$ of two classes in Figure 9d, we notice that $z = 0$ is not the optimal decision boundary anymore, because there are less negative samples than positive samples. By using negative adaptation rate $\alpha < 0$, the normalization layer can further "disalign" the intermediate feature, which can further improve the accuracy, as shown in Figure 9e.