# OpenReview forum: "Adaptive Test-Time Personalization for Federated Learning"
_NeurIPS.cc/2023/Conference — NeurIPS 2023 poster_

### Official Review · Reviewer_yJjv · 2023-06-24

**Soundness:** 3 good
**Presentation:** 3 good
**Contribution:** 3 good
**Rating:** 5
**Confidence:** 4

**Summary:**

This paper considers the federated learning setting of adaptive test-time personalization. Traditional test-time adaptation (TTA) can only handle specific target domain distributions, while federated learning requires flexible handling of multiple target domains. Existing TTA methods pre-define which modules to adapt, which limits the application of TTA in federated learning. Therefore, this paper proposes the Adaptive Test-time Personalization algorithm called ATP to automatically decide which modules to adapt and how much to adapt.

**Strengths:**

1. The experiment in Section 3.2 is crucial as it effectively illustrate the inherent challenges encountered by current TTA methods when applied to federated learning, thus offering valuable insights and guiding directions to potential enhancements.
2. The proposed method is simple and easy to implement, and it has achieved impressive performance in the current experiments.

**Weaknesses:**

1. The paper claims that it is the first to propose test-time personalized federated learning. However, this claim is questionable because previous works, such as [1], had already explored test-time personalized tasks in the context of federated learning.
[1] Jiang, Liangze, and Tao Lin. "Test-Time Robust Personalization for Federated Learning." ICLR2023 (preprint arXiv:2205.10920 (2022))
2. I am concerned about the significance of the “supervised refinement” step. If labeled data is available in this step, using the labeled data itself already leaks the distribution about the test samples in TTA tasks (while the test distribution is not known in the TTA setting), which obviously reduces the difficulty of TTA. If labeled data is not available, the proposed method for adaptively learning alpha seems unworkable.
3. The datasets used in the experiments are small-scale. It would be more convincing if experiments were conducted on real-world images, such as DomainNet, with at least a resolution of 224x224.

**Questions:**

1. In Figure 2, why does setting "bn.running_mean (m=-0.1)" improve the accuracy of TTA in the presence of label shift, while "bn.running_mean (m=0.1)" significantly harms the accuracy? This seems counterintuitive.
2. The task addressed in this paper is more akin to source-free unsupervised domain adaptation rather than Test-Time adaptation. In my opinion, the focus of most TTA works is on an online setting where the entire test set cannot be obtained at once, and operations are performed on individual test samples to be predicted (as the details in Section 4.3).
3. See Weaknesses 2, how important is the step of "supervised refinement"? Can the authors provide any ablation experiments?


**Limitations:**

There is no limitations or potential negative societal impact discussed in this paper.

---

> ### Author Rebuttal · Authors · 2023-08-09
>
> Dear Reviewer yJjv,
>
> We sincerely thank you for your comprehensive review and insightful feedback on our paper. We appreciate your positive feedback on the crucial experiment and effective algorithm of our work. We would like to address your concerns as follows.
> ## W1. Comparison to FedTHE
> > 1. The paper claims that it is the first to propose test-time personalized federated learning. However, this claim is questionable because previous works, such as [15], had already explored test-time personalized tasks in the context of federated learning.
> We would like to clarify that our proposed TTPFL, is substantially different from the setting in FedTHE [15]. FedTHE focuses on improve the model robustness to test-time shifts on seen labeled training clients (i.e., clients that participate in FL training with labeled data), while our algorithm focuses on better generalization to unseen unlabeled testing clients (i.e., clients that neither participate in training nor have labeled data) with different distributions. Our research problems are orthogonal. We discuss our differences in detail in [our response to all reviewers, part 3](https://openreview.net/forum?id=rbw9xCU6Ci&noteId=PEt62AClVa).
> ## W2. Significance of supervised refinement
> > 2. I am concerned about the significance of the “supervised refinement” step. If labeled data is available in this step, using the labeled data itself already leaks the distribution about the test samples in TTA tasks (while the test distribution is not known in the TTA setting), which obviously reduces the difficulty of TTA.
>
> Thanks for raising this concern. We would like to clarify that labeled data is only available for training clients, which participate in FL training. Neither the image nor the label for testing clients are used for supervised refinement. After training, each unseen testing client personalizes the global model in an unsupervised manner with the trained adaptation rates and test the adapted model’s performance. We report the average accuracy on testing clients.
>
> [Our response to all reviewers, part 1](https://openreview.net/forum?id=rbw9xCU6Ci&noteId=PEt62AClVa) could be helpful in further clarifying this issue.
> ## W3. Large-scale dataset
> >3. The datasets used in the experiments are small-scale. It would be more convincing if experiments were conducted on real-world images, such as DomainNet, with at least a resolution of 224x224.
>
> Thanks for your valuable advice. We further conduct experiments on PACS, a large-scale common datasets in TTA with resolution $224\times 224$. The result in the [rebuttal pdf, part B](https://openreview.net/attachment?id=PEt62AClVa&name=pdf) shows that ATP consistently outperforms all baseline.
> ## Q1. Figure 2
> > 1. In Figure 2, why does setting "bn.running_mean (m=-0.1)" improve the accuracy of TTA in the presence of label shift, while "bn.running_mean (m=0.1)" significantly harms the accuracy? This seems counterintuitive.
>
> Thanks for your insightful question. We used a visual example in Appendix C.4 to explain why different momentum m has opposite effects when adapting to label shift (m in Figure 2 refers to $\alpha$ in Appendix C.4). In brief, adapting the running mean can be seen as feature aligner/disaligner.
> - Positive m aligns the intermediate feature distributions for training and testing data, resulting in aligned distribution of predictions. However, since the label distribution is shifted, such alignment introduces negative effects.
> - Meanwhile, negative m disaligns the intermediate feature distribution, which follows the change in label distribution and improves the accuracy.
> ## Q2. Source-free unsupervised domain adaptation
> > 2. The task addressed in this paper is more akin to source-free unsupervised domain adaptation rather than Test-Time adaptation. In my opinion, the focus of most TTA works is on an online setting where the entire test set cannot be obtained at once, and operations are performed on individual test samples to be predicted (as the details in Section 4.3).
>
> Thanks for your comment. We would like to clarify that our proposed ATP does not obtain the entire test set at once. Instead,
> - For ATP-Episodic, testing clients process each test batch independently, resembling test-time batch adaptation according to [R1].
> - For ATP-Online, testing clients process test batches sequentially like Tent [39], resembling online test-time adaptation.
>
> Moreover, we would like to emphasize that our ATP is substantially different from previous works in TTA since ATP learns from multiple FL clients to tackle different types of distribution shifts.
>
> [R1] Jian Liang, Ran He, Tieniu Tan. A Comprehensive Survey on Test-Time Adaptation under Distribution Shifts.
> ## Q3. Supervised refinement
> > 3. See Weaknesses 2, how important is the step of "supervised refinement"? Can the authors provide any ablation experiments?
>
> Thanks for your question. Supervised refinement is important in ATP since it learns a set of adaptation rates specific to the type of distribution shifts among FL clients. Without supervised refinement, the adaptation rates will be initialized as zero, thus ATP becomes identical to “no adaptation”. For ablation studies, we try (1) using constant adaptation rates for all modules, and (2) fitting adaptation rates with a different meta-distribution, e.g., fitted on clients with feature shift but tested on clients with label shift. As shown in [our rebuttal pdf, Table C](https://openreview.net/attachment?id=PEt62AClVa&name=pdf),
> - When fitted on the *same type* of distribution shift, ATP significantly improves the performance on testing clients and outperforms constant adaptation rates. It verifies the importance of supervised refinement.
> - When fitted with a *different type* of distribution shift, the effectiveness of ATP noticeably decreases. It proves that supervised refinement can learn distribution-shift-specific adaptation rates, which further validate the importance of supervised refinement.

---

> > ### Comment · Reviewer_yJjv · 2023-08-18
> >
> > Thanks for the authors' response. After reading the response and the opinions of other reviewers, I still have the following questions:
> >
> > The authors claim that the testing domains considered by FedTHE have labeled data, whereas this work does not. This claim is not rigorous because, as the authors stated in their rebuttal, there are training client $i$ and test client $i$ both sampled from the same distribution $\mathcal{P}_i$, then this setting is not substantially different from the setting of FedTHE.
> >
> > What is the difference between the "unseen" clients in this paper and the typical client test set in federated learning? In the usual setup, client training and test sets also come from the same distribution, and only the training set is used during training.

---

> > > ### Author Response · Authors · 2023-08-18
> > > **Clarification of non-overlapping training and testing clients**
> > >
> > > Dear reviewer yJjv,
> > >
> > > Thanks a lot for your comment. We apologize for the confusion raised from notation of client index $i$. We would like to clarify that in our TTPFL setting, **training clients and testing clients are non-overlapping**, i.e., there are $N$ training clients $i = 1, \cdots, N$ and $M$ testing clients $j=1, \cdots, M$. **All the training and testing clients have different distributions**, i.e., $\\{\mathcal{P}\_i^{\text{train}}\\}\_{i=1}^N$ and $\\{\mathcal{P}\_j^{\text{test}}\\}\_{j=1}^M$ are all different, while these distributions are sampled from the same meta-distribution $\mathcal{Q}$, i.e., distribution of distributions.
> > >
> > > In this case, for a testing client $j$, its distribution $\mathcal{P}_j^{\text{test}}$ is different from all the training clients' distributions $\\{\mathcal{P}\_i^{\text{train}}\\}\_{i=1}^N$. Therefore, FedTHE cannot be used in our TTPFL setting since there are no labeled data from $\mathcal{P}_j^{\text{test}}$.
> > >
> > > Thanks again for your comment. We will carefully revised our manuscript to avoid confusion.

---

> > > > ### Comment · Reviewer_yJjv · 2023-08-18
> > > >
> > > > Thanks for the clarification, I see the difference in problem setting. I tend to raise my recommendation to borderline accept. Meanwhile, I suggest that the authors may re-consider the theme of this work - Compared with test time personalisation, I think this setting is more like domain generalisation.

---

### Official Review · Reviewer_2b9x · 2023-07-03

**Soundness:** 2 fair
**Presentation:** 2 fair
**Contribution:** 3 good
**Rating:** 5
**Confidence:** 4

**Summary:**

The paper studies test-time personalization in a federated learning setting --- after training on participating clients, the goal is to locally adapt the  global model given unlabeled test data. The paper's main idea is by pointing out that label non-IID and domain non-IID require adaptation on different layers of DNNs, and propose a novel way to learn the adaptation learning rates of each layer automatically in a data-driven fashion.  A simple learning method that alternatively do SGD on the DNN parameters and the layer-wise learning rates shows effective improvements in test-time personalization.

**Strengths:**

- The method proposes a novel aspect that to tailor different types of non-IIDness, the degrees of adaptation are different across layers. it is an interesting point. The solution is simple, sound, and effective.

- Empirical results are overall satisfying. The experiments provide enough comparisons to centralized TTA algorithms; see some suggestions below.


**Weaknesses:**

Although I am positive on this paper, I observe several important concerns. If the authors could address them, my score can be higher.

[Major 1] Overclaims: considering TTA in PFL setting was first introduced in [15]. This is a natural extension of centralized TTA. It is unnecessary and imprecise for this paper to make "test-time personalized federated learning (TTPFL)" a new setting. The definition in L38 about the combination of distribution shifts of labels and styles itself is nothing to do with federated setting; centralized TTA can have both label and style shifts.

[Major 2] Misleading section 4.2. I cannot understand why the proposed refinement has a connection with meta-learning. ClientTrain in Alg 1 is simply a coordinate gradient style optimization. Alternatively, a discussion about hyperparameter optimization should be more relevant.

[Major 3] Theorem 5.1 is not informative. It's simply extending the classic FedAvg generalization bound [18] for the adaptive parameters. It will be more informative and fit to the context to discuss the generalization to the new clients of different distributions ''after test-time personalization'' in TTA sense. Otherwise, I would recommend avoid this laundry theorem.

[Minor 1] The datasets are rather small scales and synthetic. it will be better to include natural federated datasets like FEMNIST or iNaturalist-GEO or large-scale common datasets in TTA (like ImageNet).

[Minor 2] Why FedTHE or FedTHE+ in [15] are not compared? [15] seems to be the closest work about TTA in FL. Is the discussion in L76 faithful? It seems to me FedTHE does not need labeled data.


**Questions:**

Section 3.2 is the limitation to TTA but not about PFL. Can the author discuss why the problem or the solution is dedicated to FL? I suggest adding a discussion around L134. In my opinion, the key is that in PFL we can figure out the ideal adaptive rates by leveraging the training clients who have diverse distributions.

**Limitations:**

See weakness.

---

> ### Author Rebuttal · Authors · 2023-08-09
>
> Dear Reviewer 2b9x,
>
> Thank you for your detailed and insightful review of our paper. We appreciate your positive feedback on the novelty and empirical results of our work. We have carefully considered your concerns and suggestions, and will address them point by point as follows.
>
> ## W1. Overclaims
>
> > [Major 1] Overclaims: considering TTA in PFL setting was first introduced in [15]. This is a natural extension of centralized TTA. It is unnecessary and imprecise for this paper to make "test-time personalized federated learning (TTPFL)" a new setting.
>
> Thanks for your question regarding the comparison of our work and FedTHE [15]. We would like to clarify that although our setting has a similar name to theirs, our problem is substantially different. FedTHE focuses on improving the model robustness against test-time shift on seen labeled training clients (i.e., clients that participate in FL training with labeled data), while our algorithm focuses on better generalization to unseen unlabeled testing clients (i.e., clients that neither participate in training nor have labeled data) with different distributions. We compare our work with FedTHE in detail in [our response to all reviewers, part 1](https://openreview.net/forum?id=rbw9xCU6Ci&noteId=PEt62AClVa).
>
> ## W2. Connection to meta-learning & hyperparameter optimization
>
> > [Major 2] Misleading section 4.2. I cannot understand why the proposed refinement has a connection with meta-learning. ClientTrain in Alg 1 is simply a coordinate gradient style optimization.
>
> Thanks for raising this confusion of the connection between supervised refinement and meta-optimization. The ClientTrain in Alg 1 shares similarity with important meta-learning algorithms, e.g., MAML [8].
> - During unsupervised adaptation, each client conducts coordinate gradient style optimization, which resembles adapting the meta-model on a specific task in MAML (line 6 in their algorithm 1).
> - During supervised refinement, each client updates the adaptation rates to minimize the post-TTA loss, which resembles updating the meta-model in MAML (line 7 in their algorithm 1). It is important to notice that in line 16 of ClientTrain, we compute the gradient w.r.t. adaptation rates $\boldsymbol{\alpha}$, not the personalized parameter $\boldsymbol{w}_{ij}$.
>
> > Alternatively, a discussion about hyperparameter optimization should be more relevant.
>
> We agree with you that a discussion about hyperparameter optimization can also be relevant. [R1] first investigated the problem of federated hyperparameter tuning and proposed FedEX that leverages weight-sharing from neural architecture search to efficiently tune hyperparameters. [R2] introduced FloRA that addresses use cases of tabular data and enables single-shot federated hyperparameter tuning. While these methods focus on improving the efficiency of hyperparameter optimization, our paper focuses on finding the optimal adaptation rates that benefit test-time personalization. We will include this discussion in the revision of our paper.
>
> [R1] Mikhail Khodak, et al. Federated Hyperparameter Tuning: Challenges, Baselines, and Connections to Weight-Sharing. NeurIPS 2021.
>
> [R2] Yi Zhou, et al. Single-shot General Hyper-parameter Optimization for Federated Learning. ICLR 2023.
>
> ## W3. Theorem 5.1
>
> > [Major 3] Theorem 5.1 is not informative. It's simply extending the classic FedAvg generalization bound [18] for the adaptive parameters. It will be more informative and fit to the context to discuss the generalization to the new clients of different distributions ''after test-time personalization'' in TTA sense. Otherwise, I would recommend avoid this laundry theorem.
>
> Thanks for your attention regarding our generalization analysis. We would like to clarify that our Theorem 5.1 actually considers the error rate **after test-time personalization**, instead of before it. As defined in Definition B.12 in Appendix B, we consider the error rate of the adapted model $\boldsymbol{w}_{ij}$, instead of the global model $\boldsymbol{w}_G$.
>
> Theorem 5.1 aims to show that if ATP achieves low post-TTA classification error over training clients after refining the adaptation rates, we are expected to get a similar low post-TTA classification error on testing clients.
>
> ## W4. More datasets
>
> > [Minor 1] The datasets are rather small scales and synthetic. it will be better to include natural federated datasets like FEMNIST or iNaturalist-GEO or large-scale common datasets in TTA (like ImageNet).
>
> Thanks for your suggestion on the improvement of our experimental setup. We further conduct experiments on PACS, a large-scale common datasets in TTA with resolution $224\times 224$. The result is shown in [the rebuttal pdf, part B](https://openreview.net/attachment?id=PEt62AClVa&name=pdf), where ATP consistently outperforms all baseline algorithms.
>
> ## W5. Comparison to FedTHE
>
> > [Minor 2] Why FedTHE or FedTHE+ in [15] are not compared? [15] seems to be the closest work about TTA in FL. Is the discussion in L76 faithful? It seems to me FedTHE does not need labeled data.
>
> Thanks for your question. We would like to clarify that we did not compare to FedTHE [15] because it requires labeled data. **FedTHE trains a model with two heads (global head and personalized head) during FL training in a supervised manner**, and fuses two heads during test-time in an unsupervised manner. When generalizing to unseen unlabeled testing clients, the client can only download the global head but cannot generate its personalized head due to the lack of labeled data. Therefore, FedTHE cannot be used for unseen unlabeled testing clients, which is the target of our paper.
>
> In our [rebuttal pdf, part A](https://openreview.net/attachment?id=PEt62AClVa&name=pdf), we design a variant of FedTHE using pseudo-labels to train the personalized head. Our algorithm outperforms this FedTHE variant across different types of distribution shifts.

---

### Official Review · Reviewer_h3BM · 2023-07-05

**Soundness:** 2 fair
**Presentation:** 2 fair
**Contribution:** 2 fair
**Rating:** 4
**Confidence:** 4

**Summary:**

This paper proposes a new setting called test-time personalized federated learning (TTPFL) and proposes an Adaptive Test-time Personalization algorithm. The authors show effectiveness of proposed method over other test-time adaptation methods.

**Strengths:**

The paper proposes an Adaptive Test-time Personalization algorithm and shows its effectiveness over other test time adaptation methods.

**Weaknesses:**

1. The proposed setting is strange and not self-consistent. The authors claim that in this setting 'clients adapt a trained global model in an unsupervised manner without requiring any labeled data.' However, the proposed method involves labeled clients to learn the learning rates for different modules (Figure 3, left). At least, real-world scenarios as examples should be provided.
2. Missing representative federated learning baselines. This paper does not compare with any federated learning method. If the proposed method uses the labeled datasets for the training stage, then many federated learning methods can be seen as baselines that are trained on the labeled datasets. Baseslines include FedAvg, FedAvg with Fine-tuning, FedProx, FedProx with finetuning, Ditto, pFedMe.
3. Since the proposed method focuses on different operations on different modules of a model, model architecture is important to this paper, thus experiments on different model architectures are required.

**Questions:**

See weakness.

**Limitations:**

Generally, I am really confused about the setting in this paper after reading. Please provide comprehensive explanations to correct me if I am wrong,  and I would consider re-rating.

---

> ### Author Rebuttal · Authors · 2023-08-09
>
> Dear Reviewer h3BM,
>
> Thank you for your insightful review and valuable feedback on our paper. We sincerely appreciate your recognition of the effectiveness of our algorithm. Regarding the weaknesses, we address them point by point as follows.
>
> ## W1. TTPFL setting
>
> > 1. The proposed setting is strange and not self-consistent. The authors claim that in this setting 'clients adapt a trained global model in an unsupervised manner without requiring any labeled data.' However, the proposed method involves labeled clients to learn the learning rates for different modules (Figure 3, left). At least, real-world scenarios as examples should be provided.
>
> Thanks for pointing out this confusion of labeled and unlabeled clients. We would like to clarify that our TTPFL setting has the same data requirement as standard FedAvg [25]. In FedAvg, the global model is trained on training clients with labeled data, and then tested on testing clients with only unlabeled data. Similarly in TTPFL, during training, the global model and the adaptation rule are optimized over labeled training clients; while during testing, each testing client downloads the global model and the adaptation rule, and personalizes the global model with only its unlabeled data. We revised the description of TTPFL according to your suggestion in [our response to all reviewers, part 1](https://openreview.net/forum?id=rbw9xCU6Ci&noteId=PEt62AClVa).
>
> ## W2. More baselines
>
> > 2. Missing representative federated learning baselines. This paper does not compare with any federated learning method. If the proposed method uses the labeled datasets for the training stage, then many federated learning methods can be seen as baselines that are trained on the labeled datasets. Baseslines include FedAvg, FedAvg with Fine-tuning, FedProx, FedProx with finetuning, Ditto, pFedMe.
>
> Thanks for your suggestion of comparing to FL baselines. We compared our algorithm to all the baselines you mentioned and summarized the experiment results in [the rebuttal pdf, part A](https://openreview.net/attachment?id=PEt62AClVa&name=pdf).
>
> We would like to clarify that although our TTPFL setting requires labeled data for training clients (same as FedAvg), it does not require any labeled data for testing clients. On the contrary, most of the existing PFL algorithms either focus on the training clients exclusively (e.g., Ditto) or require labeled data for personalization on the testing clients (e.g., Fine-tuning, pFedMe). These algorithms introduce stronger data requirements to FL systems, and cannot be used for our TTPFL.
>
> To compare to these baselines, we keep the training phase of these algorithms, while using “pseudo-labels” for personalization on testing clients according to [R1]. The pseudo-label is the prediction of the global model. Experiment results in [Table A](https://openreview.net/attachment?id=PEt62AClVa&name=pdf) shows that most FL and PFL baselines can only introduce limited improvement to the testing clients. Meanwhile, our proposed ATP-Episodic outperforms all FL baselines in the setting of TTPFL across three types of distribution shifts.
>
> [R1] Dong-Hyun Lee. Pseudo-label: The simple and efficient semi-supervised learning method for deep neural networks. Workshop on challenges in representation learning, ICML 2013.
>
> ## W3. Different model architectures
>
> > 3. Since the proposed method focuses on different operations on different modules of a model, model architecture is important to this paper, thus experiments on different model architectures are required.
>
> Thanks for your suggestion to include different model architectures. In our paper, we tried two architectures, ResNet-18 and ResNet-50, which are representative in FL [36,43,44] and TTA [14,39,45]. Experiment results show that ATP is compatible and effective with different model architectures. Please refer to Appendix C.2 for our experiments with ResNet-50.

---

> > ### Author Response · Authors · 2023-08-11
> > **Follow-up response regarding different model architectures**
> >
> > Dear Reviewer h3BM,
> >
> > Thanks again for your valuable feedback on our paper. Regarding the question of different model architecture, we would like to provide additional experiment results.
> >
> > Consider that smaller models are used in FL for their smaller communicational and computational cost, we test our algorithm with a 5-layer CNN on CIFAR-10 dataset. Other experimental settings are identical to our CIFAR-10 experiment in Section 6.
> >
> > As shown in Table D below, similar to our results in the paper, our proposed ATP achieves better performance than various baselines across  three types of distribution shifts, which further validate the effectiveness of ATP.
> >
> >
> >
> > Table D: Accuracy (%) under three kinds of distribution shifts on CIFAR-10. We report the average accuracy on testing clients.
> > | Method        | Feature shift | Label shift | Hybrid shift | Avg. Rank |
> > | ------------- | ------------- | ----------- | ------------ | --------- |
> > | No adaptation | 64.33         | 69.15       | 61.87        | 7.7       |
> > | BN            | 66.51         | 54.60       | 50.20        | 7         |
> > | SHOT          | 65.70         | 49.93       | 45.89        | 8.3       |
> > | Tent          | 65.57         | 50.02       | 45.72        | 9         |
> > | T3A           | 64.33         | 66.34       | 59.59        | 8         |
> > | MEMO          | 65.64         | 71.77       | 64.16        | 5.7       |
> > | EM            | 61.65         | *76.16*       | 67.06        | 5         |
> > | BBSE          | 56.92         | 76.13       | 66.21        | 6.3       |
> > | Surgical      | 64.45         | 73.58       | 65.35        | 5.7       |
> > | ATP-Episodic  | *66.88*         | 76.14       | *68.48*        | *2.3*      |
> > | ATP-Online    | **67.11**     | **78.38**   | **70.86**    | **1**         |

---

> > ### Comment · Reviewer_h3BM · 2023-08-20
> >
> > Thank you for the responses. However, I still have two concerns:
> >
> > (Major) W2: The authors should show the results on the training clients and compare with the FL methods in Table A. It is essential to enhance the performances of training clients to ensure the setting is practical. Specifically, the training clients need to spend computation, communication, and privacy cost (to a degree) to afford a FL system. When the proposed method improves the performance of testing clients at the cost of decreasing the performance of training clients, the training clients would refuse to participate. Therefore, it is not sufficient to only show the performance of testing clients.
> >
> > W3: The proposed method requires specific designs on convolutional neural networks with batch normalization. I'm concerned about whether the method can be effortlessly extended to other types of models, such as Transformer.

---

> > > ### Author Response · Authors · 2023-08-21
> > > **Response to Reviewer h3BM**
> > >
> > > Dear Reviewer h3BM,
> > >
> > >
> > >
> > > Thanks a lot for your suggestion. We would like to address your concerns as follows.
> > >
> > > ## W2. Comparison to FL baselines.
> > >
> > > Thanks for your comment regarding the performance on training clients. We understand your point that training clients should also benefit from the FL algorithm. According to your suggestion, we further show the results on the training clients in Table D below,  and compare our proposed ATP to FL methods. As shwon in Table D,
> > >
> > > - ATP also greatly improves the performance of training clients, compared to global FL methods (FedAvg and FedProx).
> > > - ATP has good performance on training clients, which is comparable to FL methods designed for labeled clients exclusively (FT, Ditto, pFedMe, and FedTHE).
> > >
> > >
> > >
> > > Table D. Comparison with FL baselines on CIFAR-10. We report the average accuracy (%) on *labeled training clients*.
> > >
> > > | Method       | Feature shift | Label shift | Hybrid shift |
> > > | ------------ | ------------- | ----------- | ------------ |
> > > | FedAvg       | 69.02         | 72.65       | 60.34        |
> > > | FedAvg + FT  | 69.64         | 79.38       | 70.45        |
> > > | FedProx      | 68.94         | 72.56       | 60.31        |
> > > | FedProx + FT | 69.57         | 79.47       | 70.73        |
> > > | Ditto        | 71.93         | 77.35       | **72.08**    |
> > > | pFedMe       | 61.75         | 74.91       | 71.74        |
> > > | FedTHE       | 69.95         | 79.32       | 70.32        |
> > > | ATP-Episodic | **72.17**     | **79.79**   | 70.26        |
> > >
> > >
> > >
> > > Despite achieving good performance on training clients, we still want to emphasize that **the purpose of our proposed ATP is not to maximize the effectiveness of labeled training clients**. Instead, **our goal is to achieve better adaptation and generalization to new (i.e., unparticipating) unlabeled testing clients**. In real FL systems, the ability to generalize to new clients is of paramount importance [43], as only a small fraction of clients possess labels and participate in the FL training process [16].
> > >
> > >
> > >
> > > ## W3. More architectures
> > >
> > > Thank you for bringing up this important question. We would like to clarify that our proposed algorithm doesn't involve tailored designs specifically for convolutional neural networks; rather, it adapts by learning appropriate adaptation rates for individual modules. Regarding batch normalization layer adaptation within our framework, it's essential to note that this foundational component is widely adopted in modern neural network architectures, including ResNet and newer transformer-based architectures [R2]. As a result, our ATP approach is versatile and doesn't depend on any specific backbone architecture, allowing for seamless integration onto diverse architectures.
> > >
> > > In our paper and rebuttal, we utilized ResNet-18/50 and ConvNet as backbones, in line with previous studies in FL [36,43,44] and TTA [14,22,39,45]. They have demonstrated remarkable performance on benchmark datasets we used [39,45], and they remain the most extensively explored backbone architectures in the literature [22,36,39,43,44,45], largely due to their efficiency.
> > >
> > > We genuinely appreciate your insightful suggestions, and we're enthusiastic about expanding our study to incorporate results from a broader array of backbone architectures in the future revision.
> > >
> > > [R2] Zhuliang Yao, Yue Cao, Yutong Lin, Ze Liu, Zheng Zhang, Han Hu. Leveraging Batch Normalization for Vision Transformers. ICCVW 2021.

---

> > > > ### Comment · Reviewer_h3BM · 2023-08-21
> > > >
> > > > * I recognize that the primary objective is to enhance generalization to unlabeled testing clients. However, if the approach doesn't first benefit labeled training clients, it would be an impractical setting. From the authors' feedback, it appears this concern wasn't sufficiently addressed.
> > > >
> > > > * The experiments on the training clients are helpful. However, except for Feature shift, the other two do not improve much.
> > > >
> > > > After much consideration, I've improved my score, but I still believe this paper requires further improvement.

---

### Official Review · Reviewer_A9BH · 2023-07-08

**Soundness:** 3 good
**Presentation:** 3 good
**Contribution:** 2 fair
**Rating:** 5
**Confidence:** 4

**Summary:**

This paper introduces a novel setting where personalized FL during the test procedure is considered and multiple distribution shifts are involved. A method termed ATP is proposed to solve the challenges posed in this setting. Adaptive learning rates are learned for the model. Both theoretical and empirical studies are carried out to demonstrate the effectiveness of ATP.

**Strengths:**

1. The paper is well-organized and easy to follow.

2. The main claims, e.g., the capability of dealing with multiple distribution shifts and the effectiveness of adaptive learning schemes, are well supported by empirical studies.

3. Theoretical analyses on both the convergence and generalization ability of ATP are provided.

**Weaknesses:**

1. One of the key factors of TTPFL is confusing. Usually, we assume test data are unlabeled and the target of the classification task is to predict the label of the test data. However, in the summarization of TTPFL, it is emphasized that each testing client only has unlabeled data for personalization. It is better to further clarify this factor.

2. The relation between FL and test-time shift is weak. It seems that the proposed adapting trainable parameters, adapting running statistics, and adapting rates can also benefit centralized test-time shift problems. For FL systems, the unique challenges brought by test-time shift issues and how they motivate these adapting solutions are not explicitly demonstrated.

3. Lack of detailed discussion on the difference between this work and the previous study[15]. [15] also considers feature shift, label shift, and a mixture of these shifts in their recent paper. There are various feature shifts in [15], which can also be considered as part of the experimental setting in this work.

**Questions:**

In Section 4.2, the authors have mentioned the smaller communication costs achieved by ATP, are there any corresponding experimental results that demonstrate this in a quantitive manner?

**Limitations:**

This paper does not adequately address the limitations in terms of privacy issues, efficiency, etc.

---

> ### Author Rebuttal · Authors · 2023-08-09
>
> Dear Reviewer A9BH,
>
> We would like to express our sincere gratitude for the thoughtful review and constructive feedback provided. We are grateful for your positive feedback on the paper's organization, empirical support, and theoretical analyses. We have carefully considered your comments and will address each concern in this rebuttal.
>
> ## W1. TTPFL setting
>
> > 1. One of the key factors of TTPFL is confusing. Usually, we assume test data are unlabeled and the target of the classification task is to predict the label of the test data. However, in the summarization of TTPFL, it is emphasized that each testing client only has unlabeled data for personalization. It is better to further clarify this factor.
>
> Thank you for pointing out this confusion. We emphasized that each testing client only has unlabeled data for personalization, in comparison to traditional PFL algorithms like Per-FedAvg [7], pFedMe [6], and FedTHE [15]. In these algorithms, testing clients have both labeled data (for personalization) and unlabeled data (for evaluation/prediction), which is a stronger requirement than ours. However, in the test-time of TTPFL, our ATP algorithm only requires unlabeled data for local personalization, utilizing the distribution of unlabeled data to adapt the global model and achieve better performance. We revised the description of TTPFL according to your suggestion in [our response to all reviewers, part 1](https://openreview.net/forum?id=rbw9xCU6Ci&noteId=PEt62AClVa).
>
> ## W2. Relation between FL and test-time shift
>
> > 2. The relation between FL and test-time shift is weak. It seems that the proposed adapting trainable parameters, adapting running statistics, and adapting rates can also benefit centralized test-time shift problems. For FL systems, the unique challenges brought by test-time shift issues and how they motivate these adapting solutions are not explicitly demonstrated.
>
> Thank you for pointing out this issue. We summarize the unique challenges of TTPFL in comparison with centralized TTA as follows:
> 1. **Various distribution shifts**: Since each FL client collects its data in a distributed manner, the data can exhibit multiple distribution shifts, e.g., a complex combination of feature and label shifts. However, the complexity of distribution shifts have been overlooked in most centralized TTA works. As shown in Subsection 3.2, *most TTA algorithms cannot tackle feature and label shifts simultaneously*. However, our ATP algorithm can learn to tackle different types of distribution shifts as shown in Subsection 6.1.
> 2. **Exploiting multiple data sources**: *centralized TTA considers adaptation from one source domain to one target domain, while our TTPFL includes multiple clients as data sources*. In TTPFL, centralized TTA algorithms adapt a global model which can only exploit training clients’ data as a mixed distribution $\mathcal{P}_G = \sum_i p_i \mathcal{P}_i$, ignoring how they are different from each other. However, our ATP algorithm learns how each client’s distribution $\mathcal{P}_i$ is different from the mixed distribution $\mathcal{P}_G$, and optimizes the adaptation rates accordingly.
>
> ## W3. Discussion of FedTHE
>
> > 3. Lack of detailed discussion on the difference between this work and the previous study[15]. [15] also considers feature shift, label shift, and a mixture of these shifts in their recent paper. There are various feature shifts in [15], which can also be considered as part of the experimental setting in this work.
>
> Thanks for your suggestion for a detailed comparison with FedTHE [15]. Our paper is substantially different from FedTHE [15] in research problems, algorithms, and experiment design. We discuss our differences in detail in [our response to all reviewers, part 3](https://openreview.net/forum?id=rbw9xCU6Ci&noteId=PEt62AClVa). Specifically, regarding the experiment design, while FedTHE and our paper use similar techniques to construct feature and label shifts, we consider a stronger fusion of them. In FedTHE, the author simply mixed samples from different shifted distributions, which mitigates the severity of distribution shifts. For example, a smaller portion of testing samples are corrupted. As a result, “mixture of test” is less challenging than “corrupted local test” as shown in Table 1 of [15]. Noticing this deficiency, we improved the way of mixing feature and label shifts in our paper, making “hybrid shift” more challenging than both feature and label shifts (as shown in Table 1 of our paper).
>
> ## Q1. Communication cost
>
> > In Section 4.2, the authors have mentioned the smaller communication costs achieved by ATP, are there any corresponding experimental results that demonstrate this in a quantitive manner?
>
> Thanks for mentioning the advantages of ATP in communication efficiency. Unlike FedAvg [25] which transmits the full model parameters in every communication round, training clients in ATP only download the full model once, and only transmit the adaptation rates during the training process. Therefore, for $T$ communication rounds, ATP only transmits $D+2Td$ floating numbers, much fewer than $2TD$ for FedAvg. Notice that $d \ll D$: For our ResNet-18 experiments on CIFAR-10, $d=102$ while $D=11,181,642$. For our ResNet-50 experiments on CIFAR-100, $d=267$ while $D=23,581,642$.
>
> ## Limitations
>
> > This paper does not adequately address the limitations in terms of privacy issues, efficiency, etc.
>
> We appreciate your comment. Regarding privacy, while our ATP framework shares the same communication protocol as FedAvg while transmitting fewer parameters, we did not further study the privacy of our framework. We believe differential privacy techniques can further protect the privacy of clients in ATP systems.

---

> > ### Comment · Reviewer_A9BH · 2023-08-21
> > **Thanks for rebuttal.**
> >
> > Hi Authors, Thanks for your rebuttal that partially solved my concern. I will raise my score to 5.

---

### Author Rebuttal · Authors · 2023-08-09

Dear Reviewers,

We would like to express our sincerest gratitude for your thoughtful and insightful reviews of our paper. We are particularly grateful for the recognitions bestowed upon our work, including
- novel and effective algorithm (from all reviewers),
- satisfying experiments (from reviewer A9BH, 2b9x, yJjv), and
- comprehensive convergence and generalization analysis (from reviewer A9BH).

Your insights have been immensely helpful in refining our work. In this response, we address the common questions raised by your reviews.

## 1. Revised description of test-time personalized federated learning (TTPFL)
We consider an FL system with $N$ training clients (who participate in FL training, i.e., "seen" during training) and $M$ testing clients (who do not participate, i.e., "unseen"). Each training client $i$ has its labeled dataset $S_i$ with $m_i$ i.i.d. data points drawn from its own underlying distribution $\mathcal{P}_i$, while each testing client $i$ has only unlabeled data drawn from its distribution $\mathcal{P}_i$. The data distribution $\mathcal{P}_i$ is usually different across the clients, and is sampled from a meta-distribution $\mathcal{Q}$, i.e., a distribution of distributions. **Our data requirement is identical to FedAvg [25] and FedSR [26].**

During FL training, the global model $w_G$ and the adaptation rule $\mathcal{A}$ (i.e., adaptation rates in ATP) are optimized only over labeled training clients. Testing clients’ data is not used during training. During FL testing, each testing client downloads the global model and the adaptation rule, and personalizes the global model with only its unlabeled data. The testing clients’ labels are never used in FL training or testing.

It is important to notice that **our TTPFL setting does not require any labeled data for testing clients**, unlike the setting in traditional personalized federated learning (PFL). Most of the existing PFL algorithms either focus on the training clients exclusively [35,19] or require labeled data for personalization on the testing clients [7,5]. These algorithms introduce stronger data requirements to FL systems, and cannot be used for our TTPFL.

Considering the enormous number of testing clients (compared to training clients) in real FL applications [16] and the challenge of generalization to testing clients [43], we pay special attention to the performance on the testing clients, and **report the average accuracy on testing clients**, instead of training clients (as in traditional PFL algorithms).

## 2. Relation between FL and test-time adaptation (difference between TTPFL and TTA)
Different from centralized TTA, TTPFL has its own challenges, which TTA algorithms do not fully tackle.
1. **Various distribution shifts**: Since each FL client collects its data in a distributed manner, the data can exhibit multiple distribution shifts, e.g., a complex combination of feature and label shifts. However, the complexity of distribution shifts have been overlooked in most centralized TTA works. As shown in Subsection 3.2, *most TTA algorithms cannot tackle feature and label shifts simultaneously*. However, our ATP algorithm can learn to tackle different types of distribution shifts as shown in Subsection 6.1.
2. **Exploiting multiple data sources**: *centralized TTA considers adaptation from one source domain to one target domain, while our TTPFL includes multiple clients as data sources*. In TTPFL, centralized TTA algorithms adapt a global model which can only exploit training clients’ data as a mixed distribution $\mathcal{P}_G = \sum_i p_i \mathcal{P}_i$, ignoring how they are different from each other. However, our ATP algorithm learns how each client’s distribution $\mathcal{P}_i$ is different from the mixed distribution $\mathcal{P}_G$, and optimizes the adaptation rates accordingly.

## 3. Difference to FedTHE/FedTHE+ [15]
Despite having a similar name, our paper tackles a problem substantially different from FedTHE.
- **Problem**: FedTHE focuses on improving the model robustness to test-time shift on seen labeled training clients (i.e., clients that participate in FL training with labeled data), while our algorithm focuses on better generalization to unseen unlabeled testing clients (i.e., clients that neither participate in training nor have labeled data) with different distributions. Our research problems are orthogonal.
- **Algorithm**: FedTHE trains a model with two heads (global head and personalized head) during FL training in a supervised manner, and fuses two heads during test-time in an unsupervised manner. When generalizing to unseen unlabeled testing clients, the client can only download the global head but cannot generate its personalized head due to the lack of labeled data. Therefore, FedTHE algorithm cannot be used for unseen unlabeled testing clients, which is the target of our paper. Different from FedTHE, testing clients in ATP download the global model as well as the adaptation rates, and adapt the global model locally with its unlabeled data. Labels are not required in ATP testing.
- **Experiment**: While FedTHE and our paper use similar techniques to construct feature and label shifts, we consider a stronger fusion of them. In FedTHE, the author simply mixed samples from different shifted distributions, which mitigates the severity of distribution shifts. For example, a smaller portion of testing samples are corrupted. As a result, “mixture of test” is less challenging than “corrupted local test” as shown in Table 1 of [15]. Noticing this deficiency, we improved the way of mixing feature and label shifts in our paper, making “hybrid shift” more challenging than both feature and label shifts (as shown in Table 1 of our paper).

## 4. New experiments
In the rebuttal pdf, we conduct experiments regarding
1. comparison to FL baselines (including FedTHE)
2. new experiments on PACS, a large-scale dataset, and
3. additional ablation study.

Details are provided in the pdf.

---

### Author Response · Authors · 2023-08-15
**Gentle Reminder**

Dear Reviewers,

Thank you very much for your time and efforts on our paper. We are writing to follow up on our rebuttal. We have carefully considered the raised questions and made our responses as below. As the discussion period is halfway through, we gently ask for a retrospect regarding our rebuttal and whether there are any remaining questions or concerns that you would like to address.

Thanks again for your time and consideration!

---

> ### Author Response · Authors · 2023-08-19
>
> Dear Reviewers,
>
> As the author-reviewer discussion phase is drawing to a close, we respectfully ask for your confirmation of our replies. Please inform us if there are any remaining comments that you feel have not been adequately addressed. Additionally, we would be grateful if you could verify whether your evaluation aligns with your revised comprehension of our research.
>
> We sincerely appreciate your time and consideration.

---

### Decision · Program_Chairs · 2023-09-21

**Decision:**

Accept (poster)

**Comment:**

Initially, this submission received mixed scores. Several reviewers were confused about the problem setting and the proposed solution. The authors addressed this, as well as concerns about the similarity to prior work, runtime and data usage. On the positive side, the submission provides a new solution to a relevant problem, which is backed up by extensive experimental evaluation as well as some theoretical results about generalization as well as convergence. Ultimately, a majority of the reviewers recommended accepting the work.